# Giant iceberg behaviour impacts regional biogeochemical cycling in the Southern Ocean
Laura R. Taylor [1,2] ✉, Helena Pryer[2], Katharine R. Hendry [1], Rachael N. C. Sanders [1], Michael P. Meredith [1,6], Andrew Meijers [1,6], Edward Mawji [3,6], E. Malcolm S. Woodward [4,6], Carol Arrowsmith[5,6], Melanie J. Leng[5,6], E. Povl Abrahamsen [1], Helen M. Williams [2] & Clara Manno [1] ✉

Giant Antarctic iceberg calving is projected to increase with climate change, affecting ocean circulation, nutrient supply, and carbon cycling. These icebergs can stimulate primary production and influence Southern Ocean carbon fluxes through modification of upper ocean physics and biogeochemistry, yet the underlying mechanisms remain poorly constrained. We investigate the coupled effects of meltwater input and nutrient dynamics around two of the largest known icebergs, A-76A and A-23A, using silicon isotopes alongside hydrographic, meltwater, and macronutrient observations to examine nutrient cycling. Around A-76A, enhanced glacial meltwater input coincides with macronutrient variability and strong silicon isotope fractionation, indicating diatom utilisation sustained by continued macronutrient supply. In contrast, waters around A-23A show minimal glacial meltwater enhancement and remain macronutrient-rich, with no silicon isotope fractionation, indicating limited biological uptake despite favourable background conditions. These contrasting regimes reveal that iceberg influence on ocean biogeochemistry is highly heterogeneous, reflecting the combined effects of micronutrient fertilisation, macronutrient resupply, and environmental context. Our findings demonstrate that giant icebergs exert dual controls on productivity by initiating blooms through micronutrient delivery and sustaining biomass accumulation through resupply from depth. This mechanistic understanding is critical for assessing the role of increasing iceberg discharge in future Southern Ocean productivity and carbon cycling.

Anthropogenic climate change is driving rapid transformations across the Earth's cryosphere, with accelerating ice-sheet loss exerting far-reaching impacts on sea level, ocean circulation, and global biogeochemical cycles. The West Antarctic Ice Sheet (WAIS) is becoming increasingly unstable, with mass loss projected to continue throughout the 21st century[1]. As a result, the discharge of giant icebergs into the Southern Ocean is expected to increase[2]. These icebergs—defined as exceeding 18 km in length[2]—strongly influence ocean circulation, nutrient supply, and marine biodiversity, with cascading impacts on global climate systems[3,4]. By stimulating primary production and enhancing organic carbon export, they may account for 10–20% of Southern Ocean carbon flux[4]. Yet, the mechanisms through

which iceberg passage enhances productivity—particularly the processes that control the magnitude and persistence of biological responses—remain poorly constrained, limiting our ability to predict the consequences of accelerating WAIS-derived iceberg discharge for carbon cycling and ecosystem dynamics in the Southern Ocean.

Observations of iceberg-induced productivity enhancement are highly variable, with reported increases in chlorophyll-$\alpha$ (chl-$\alpha$) concentrations differing across regions[5,6], iceberg properties[6], and between in situ and remotely sensed observations[4]. This variability complicates quantitative estimation of iceberg-driven productivity and hinders assessment of their contribution to Southern Ocean carbon cycling. At large scales, satellite-

[1]British Antarctic Survey, Cambridge, United Kingdom. [2]Department of Earth Sciences, University of Cambridge, Cambridge, United Kingdom. [3]National Oceanography Centre, Southampton, United Kingdom. [4]Plymouth Marine Laboratory, Plymouth, United Kingdom. [5]British Geological Survey, Keyworth, United Kingdom. [6]These authors contributed equally: Michael P. Meredith, Andrew Meijers, Edward Mawji, E. Malcolm S. Woodward, Carol Arrowsmith, Melanie J. Leng. ✉e-mail: laulor77@bas.ac.uk; clanno@bas.ac.uk

derived observations typically show increases in chl-$\alpha$ ranging from modest enhancements to a tenfold increase above background concentrations[4–6]. At smaller spatial and temporal scales, however, in situ studies reveal a more complex picture: there is a lag time between iceberg passage and productivity enhancement[5,7–10], chl-$\alpha$ concentrations may be decoupled from net community production[10], and productivity maxima can occur at a ranges of distances from the iceberg due to local dilution and advection[7–9,11]. Regional oceanography and sea ice dynamics also modulate these effects: in the Ross Sea, located in the Pacific sector of the Southern Ocean, giant icebergs suppressed productivity by increasing sea ice cover[12] and delaying the spring bloom[13]. Even palaeoceanographic records show conflicting signals of diatom productivity during periods of intense iceberg discharge[14,15]. Together, these observations suggest that iceberg-associated productivity responses are governed by interacting physical, chemical, and biological controls, rather than a single dominant fertilisation mechanism.

One factor underlying this variability is the nature of nutrients released as icebergs melt, particularly the delivery of iron and other micronutrients that alleviate iron limitation and enable primary production in the Southern Ocean. The dissolution of terrigenous material entrained within glacial ice represents a potentially important source of micronutrients[11,16,17], particularly bioavailable iron and manganese[18,19]; however, icebergs are a negligible source of macronutrients, and meltwater may dilute baseline concentrations[20]. Short-term differences in primary production perturbation have been linked to the geographic origin of icebergs, potentially reflecting contrasts in the geology of their source catchments[4]. However, although the composition of terrigenous material can vary substantially within a single iceberg, no systematic differences have been identified between Antarctic sectors despite distinct regional lithologies[20]. This suggests that compositional heterogeneity alone cannot explain the observed variability in productivity responses. While micronutrient fertilisation is therefore a likely prerequisite for iceberg-induced productivity enhancement in much of the Southern Ocean, it does not fully account for the wide range of observed response magnitudes.

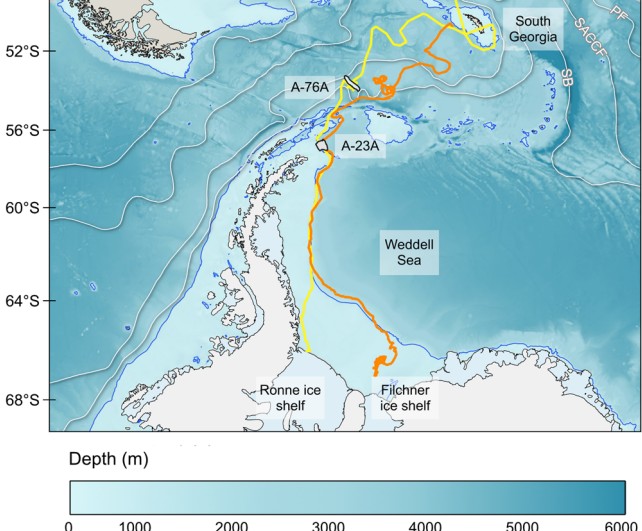

**Fig. 1 | Trajectories of giant icebergs A-76A and A-23A across the Southern Ocean.** Outlines show iceberg positions and to-scale size on the respective sampling days. The track of iceberg A-76A is shown in yellow and A-23A in orange. The mean positions of the major fronts of the Antarctic Circumpolar Current (ACC)[73] are shown in white: the Polar Front (PF), Southern ACC Front (SACCF), and the Southern Boundary of the ACC (SB). Bathymetry from IBCSO v2[74] is shaded, with the 1000 m isobath highlighted in blue. Coastline and ice-shelf data are from the SCAR Antarctic Digital Database (accessed 2025). Iceberg paths and positions were generated by the British Antarctic Survey Mapping and Geographical Information Centre, using data from US NSIDC iceberg tracking products.

In addition to micronutrient release, iceberg melting can stimulate nutrient delivery through upwelling. The balance between physical melt processes governs the extent of this nutrient transport[21]. Three principal mechanisms are recognised: (1) buoyant upwelling of meltwater along the ice wall without mixing; (2) sidewall melting with ambient seawater entrainment, which drives vertical mixing; and (3) stratified melting along isopycnals, with limited vertical exchange[9,21]. Buoyant upwelling can entrain nutrient-rich circumpolar deep water (CDW) into the surface layer because the submerged drafts of giant icebergs often penetrate below the mixed layer, supplementing the nutrients released in meltwater[21]. Such macronutrient resupply may be necessary to sustain growth initiated by micronutrient fertilisation and to allow blooms to develop to high magnitudes. The relative contribution of direct surface fertilisation versus vertical mixing and deep-water entrainment varies among icebergs, which may explain observed differences in nutrient delivery and the potential for enhanced productivity[21,22].

Despite advances in understanding the physical and biogeochemical effects of iceberg melt, the mechanisms driving variability in their influence on surface ocean productivity remain unresolved. In particular, it remains unclear how micronutrient fertilisation, macronutrient resupply, and background environmental conditions interact to control the magnitude and expression of iceberg-associated biological responses. Although existing research has focused primarily on micronutrient fertilisation, the role of icebergs in modulating macronutrient distributions and stoichiometry through freshwater input and upwelling processes remains poorly understood, particularly in relation to sustaining high-magnitude phytoplankton blooms over time. Resolving these processes is essential for predicting how increasing iceberg discharge will alter Southern Ocean biogeochemistry and carbon cycling.

To address the question of what controls the magnitude and expression of primary productivity responses to giant iceberg passage in the Southern Ocean, we contrast two giant icebergs with distinct histories and regional settings, A-76A and A-23A, to examine when iceberg passage initiates, sustains, or fails to generate a biological response. We integrate macronutrient stoichiometry, stable silicon isotope composition, and freshwater fraction analyses to characterise how each iceberg modifies its surrounding water mass and phytoplankton community structure. This approach allows us to evaluate how macronutrient resupply and environmental preconditioning, in the context of previously document limiting nutrient fertilisation, combine to produce contrasting productivity responses between different icebergs in different oceanographic settings.

## Results and discussion
### Trajectories and evolution of A-76A and A-23A
To interpret the biogeochemical observations that follow, we first summarise the trajectories and environmental histories of the two icebergs studied, A-76A and A-23A. Understanding the timing and location of sampling relative to each iceberg's life cycle is critical for evaluating how their physical and chemical influences interact with background environmental conditions to control the magnitude and expression of primary productivity. Both observations were made during intermediate stages of the icebergs' lifetimes, after initial calving but prior to advanced fragmentation or decay.

Iceberg A-76 calved from the western Ronne Ice Shelf on 13th May 2021 with an area of ~4320 km², initially the world's largest iceberg. It soon broke into three fragments, of which A-76A (~3800 km²) drifted northward through the Weddell Sea and into the Antarctic Circumpolar Current (ACC) along the route known as "iceberg alley"[23] as shown in Fig. 1. Our sampling took place in early 2023, roughly 18 months after calving, when A-76A remained close to its initial ACC entry point. For the preceding ~3.5 months, it had rotated in a quasi-stationary position, likely due to a strong meander in the southern boundary of the ACC[24]. This extended residence may have facilitated repeated cycles of micronutrient release in meltwater, potential macronutrient entrainment via buoyant upwelling, and subsequent utilisation by phytoplankton, providing an ideal case study for

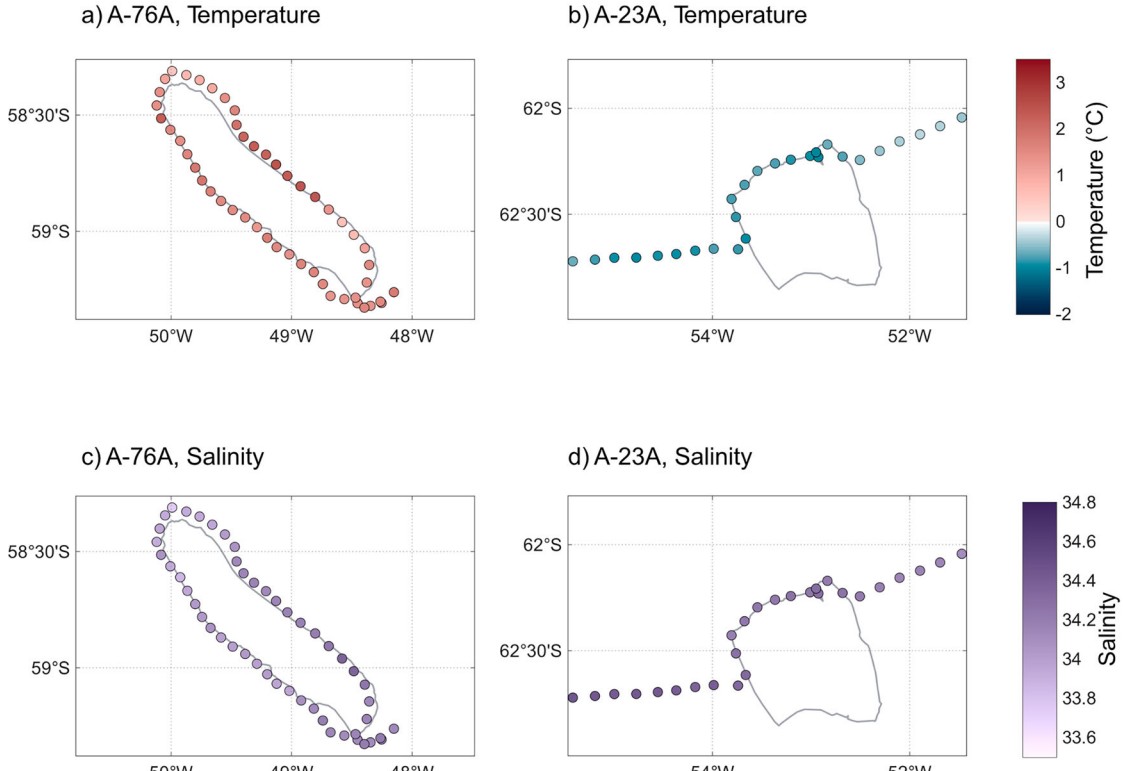

**Fig. 2 | Temperature and salinity in the vicinity of giant icebergs A-76A and A-23A.** Panels show temperature around A-76A (**a**) and A-23A (**b**) and salinity around A-76A (**c**) and A-23A (**d**).

examining sustained, high-magnitude productivity enhancement. After sampling, A-76A continued northeastward and disintegrated close to South Georgia in May 2023.

Iceberg A-23 calved from the Filchner Ice Shelf in 1986, and its largest fragment, A-23A, soon grounded on the Weddell Sea continental shelf for more than 30 years. From 2020, the ~4000 km² iceberg began intermittently moving northwestward, entering the ACC in early 2024 as shown in Fig. 1. Sampling occurred in late 2023, near the northern edge of the Antarctic Peninsula in Powell Basin. Following our observations, A-23A became temporarily trapped in a Taylor Column over Pirie Seabank in August 2024 before drifting onwards to ground, and begin disintegrating, near South Georgia in March 2025.

Satellite imagery indicates minimal change in A-23A's area from shortly after calving to when it became fully grounded (1987–1989: 5510 km²; Fig. 1). However, the iceberg lost ~25% of its area during the grounded period (2020: 4120 km²), with further reduction as the iceberg moved northwards until the day of sampling (3940 km²). This substantial reduction likely reflects decades of exposure to the ocean and gradual loss of surface sediments. While no studies have examined sediment loss in grounded, decaying giant icebergs, those that drift slowly from sectors B (90–180° W) and C (90–180° E) of the Southern Ocean toward the ACC are hypothesised to lose surface sediment load en route, reducing the micronutrient content of their meltwater and limiting downstream biological response[4,25]. A-23A's decades-long grounding could have similarly depleted surface sediments, reducing micronutrient fertilisation potential and, consequently, its capacity to drive sustained productivity.

Temporal dynamics of iceberg-ocean interactions influence how these contrasting histories manifest biogeochemically. Meltwater enters the water column immediately upon release, yet chl-$\alpha$ responses typically lag by 36 h[7] to 10 days[8], and physical restructuring of the upper ocean can persist for up to 20 days[9]. A-23A remained within roughly 20 km of the sampling site for about 5 days before and after our observations. This window may have been sufficient to detect initial productivity responses but may have been too

short to capture a fully developed bloom, particularly if background environmental conditions were limiting. In contrast, A-76A's quasi-stationary position over several months provides ample time for detecting an iceberg-induced bloom, although the iceberg's rotation complicates the spatial attribution of productivity signals.

While the exact draft and submerged geometry of the icebergs were not measured during the cruises, we assume that both icebergs were sufficiently deep to penetrate below the mixed layer, consistent with previous studies of giant icebergs[7,21,25]. Under this assumption, both icebergs had the potential to induce buoyant upwelling and entrainment of nutrient-rich CDW, a key subsurface macronutrient reservoir in the region, providing a pathway for macronutrient resupply that could support sustained phytoplankton growth where permitted by micronutrient availability.

Overall, the contrasting ages, movement patterns, and environmental exposures of A-76A and A-23A provide context for interpreting differences in meltwater delivery, nutrient supply, and biological response. A-76A provides a case in which both micronutrient fertilisation and potential macronutrient resupply are plausible, supporting high-magnitude, sustained productivity, whereas A-23A represents a scenario where decades of grounding and gradual erosion may have depleted sediment layers, limiting its biogeochemical influence. These patterns highlight how iceberg history can contribute to surface ocean biogeochemistry, providing context for analysis of nutrient dynamics and productivity responses.

### Environmental contexts shape iceberg biogeochemical influence

To assess how iceberg passage interacts with local oceanographic conditions to modulate productivity, we compared the physical and biological signatures around icebergs A-76A and A-23A. This comparison highlights the role of pre-existing environmental context in shaping the magnitude of iceberg-associated phytoplankton responses.

Salinity was lower near A-76A compared to A-23A (33.78–34.36 vs. 34.11–34.45; two-samples $t$-test; $p < 0.001$; Fig. 2). Surface salinities around

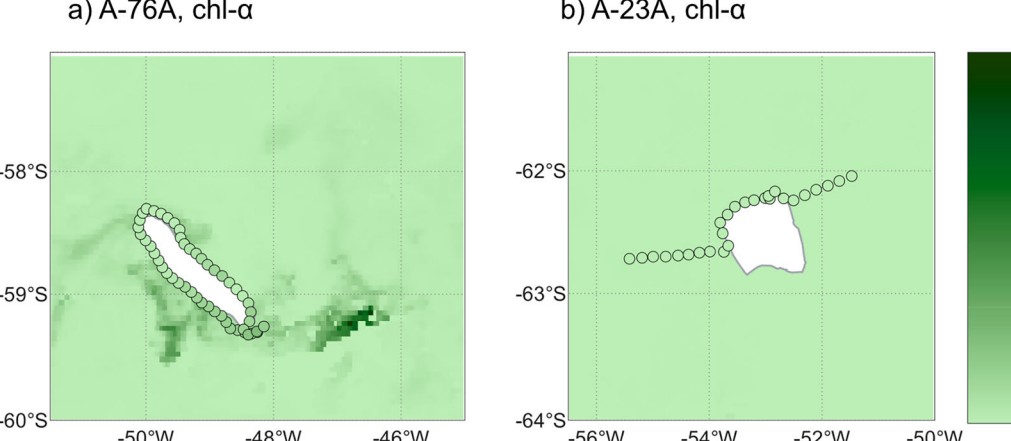

**Fig. 3 | Satellite and ship-board chlorophyll-α concentrations surrounding giant icebergs A-76A and A-23A.** Eight-day mean satellite chl-α concentration (mg m⁻³) from Aqua-MODIS (4 km resolution;[32]) surrounding icebergs A-76A (**a**; 25 January - 1 February 2023), and A-23A (**b**; 3-10 December 2023. Discrete points show ship-board continuous underway chl-α concentrations averaged to 5-min intervals.

A-23A are comparable to the World Ocean Atlas 2023 (WOA23) seasonal climatology for this area ($\approx$33.8–34.2[26]), whereas surface waters near A-76A are slightly fresher than climatological values ($\approx$33.9–34.2[26]) at the northeastern flank of the iceberg. This pattern is consistent with surface freshening due to meltwater input from iceberg A-76A; however, pronounced background variability in sea surface salinity within the ACC frontal zone complicates attribution of this signal directly to iceberg-driven processes. These observations therefore indicate potential, but not definitive, freshwater modification of the surface layer.

Sea surface temperature (SST) was higher near A-76A (0.5–2.46 °C) than A-23A ($-1.03$ to $-0.32$ °C; $p < 0.001$; Fig. 2), broadly in line with their respective regional settings, although both slightly cooler than the WOA23 seasonal climatologies for their areas (A-76A $\approx$0.8–2.8 °C; A-23A $\approx-0.7$–0.6 °C[27]. Ocean temperature plays a key role in iceberg melting; however, SST is not the dominant driver of giant iceberg melt. Most melting results from interactions with warm subsurface water masses[28], meaning iceberg draft controls melt potential[29]. Velocity shear and fragmentation also modulate melt rate, introducing variability that cannot be resolved without concurrent measurements of draft, subsurface temperature, and flow fields[29–31]. As these data were unavailable during our study, we avoid attributing melt signatures to surface temperature; consequently, the observed contrast between A-76A and A-23A cannot be explained by SST-driven melt alone.

Mean ship-board chl-α was 8.5 times higher around A-76A (3.32 mg m⁻³) than A-23A (0.39 mg m⁻³; $p < 0.001$; Fig. 3), with a peak of 10.3 mg m⁻³ at A-76A's southern flank. Satellite observations show phytoplankton blooms generally peak 50–200 km from giant icebergs[4]. Around A-23A, there are no identifiable peaks in chl-α at a distance from the iceberg (Fig. 3). However, Around A-76A, they reveal chl-α maxima of 25–41 mg m⁻³—up to four times higher than in situ values—and extending roughly 100 km from the iceberg (Fig. 3). Ship-board sensors captured the near-field signal, at a concentration approximately half of the satellite imagery; however, the ship's track missed the more distant maxima evident in Fig. 3. The differences in chl-α in the waters surrounding the two icebergs are somewhat consistent with broader regional contrasts— chl-α around A-23A is comparable to the NASA Aqua-MODIS seasonal summer climatology for the area ($\approx$0.20 mg m⁻³[32]). While the region of A-76A typically has higher chl-α concentrations (up to $\approx$0.37 mg m⁻³[32]), the magnitude observed around the iceberg far surpasses typical values, suggesting a likely iceberg-induced enhancement of productivity.

A previous study examining the influence of background chl-α concentrations on iceberg-induced phytoplankton growth found that icebergs transiting regions of relatively high chl-α typically induce a decrease in surface chl-α immediately following passage, likely due to dilution by meltwater. In contrast, icebergs moving through low chl-α regions were more likely to generate short-term increases in surface chl-α[5]. In the present study, however, iceberg A-76A, which transited waters with higher background chl-α, exhibits a clear enhancement in productivity, whereas no comparable response is observed around A-23A. Productivity increases associated with iceberg passage are expected only when the iceberg supplies a nutrient that is limiting phytoplankton growth. Given the elevated productivity of the ACC frontal region, it is likely that macro- and micro-nutrients have been depleted by prior phytoplankton growth, such that iceberg-derived nutrient inputs act to sustain biomass production. In contrast, the lower background productivity in the region surrounding A-23A may reflect alternative growth limitations, including micronutrient (e.g. iron limitation) or light limitation arising from deep mixed layers or recent sea ice cover. Nonetheless, previous studies have documented enhanced primary productivity associated with the passage of giant icebergs in the northwest Weddell Sea[5,10,11], indicating that regional background productivity is insufficient to fully explain the absence of a detectable productivity response around A-23A during our sampling period.

Together, these observations suggest that the pronounced productivity enhancement near A-76A reflects the interplay between iceberg-derived perturbations and a pre-conditioned surface ocean state, whereas the lack of a comparable response around A-23A highlights how background physical and biogeochemical conditions can constrain the expression of iceberg-driven productivity enhancement.

**Meltwater fluxes shape local hydrography**

Understanding the distribution of iceberg-derived meltwater is critical to evaluating how icebergs like A-76A and A-23A influence local nutrient dynamics and potential productivity enhancements. Freshwater provenance was assessed using concurrent salinity and oxygen isotope composition, $\delta^{18}O$, measurements, allowing separation of meteoric (glacial and precipitation) and sea ice contributions to the surface ocean. Measured $\delta^{18}O$ is sensitive to polar precipitation and glacial melt, while sea ice processes primarily affect salinity, enabling quantification of the fractional contributions of meteoric water ($F_{met}$) and sea ice melt ($F_{sim}$) to surface freshwater (see Methods)[33,34]. Freshwater fractions are derived within a closed three-endmember mass balance, meaning that increases in one freshwater component necessarily reduce the contribution of one or more other endmembers. In this study, the quantification of sea ice melt is included to separate meteoric freshwater from other freshwater sources, rather than to investigate sea ice melt or formation processes in their own right (see S2).

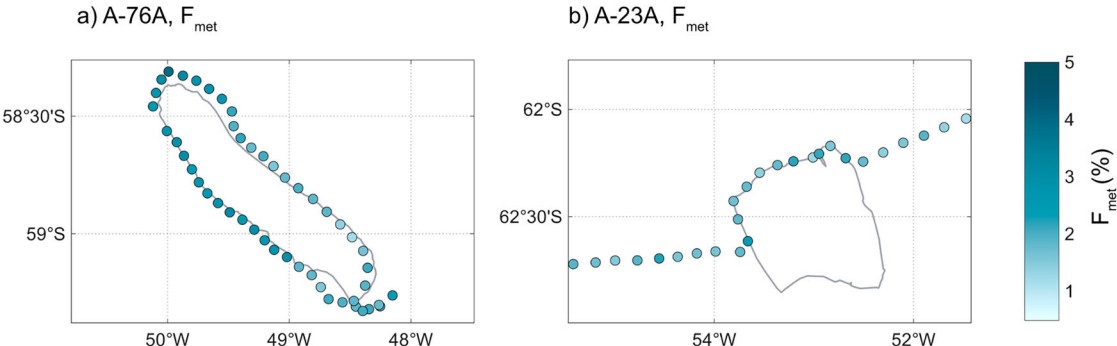

**Fig. 4 | Meteoric water fraction distributions surrounding giant icebergs A-76A and A-23A.** Panels show the meteoric water fraction ($F_{met}$) in the vicinity of icebergs A-76A (**a**) and A-23A (**b**).

Around A-76A, $\delta^{18}O$ values were significantly lower than around A-23A (−0.64 to −0.12‰ vs. −0.35 to −0.10‰; two-samples t-test; $p < 0.001$), accompanied by lower salinity, shifting surface waters towards the meteoric endmember (Supplementary Fig. 3). Correspondingly, $F_{met}$ was elevated (1.20–3.94%) relative to the 1–2% background range typical of the pelagic Southern Ocean[35,36], with the highest contributions at the iceberg's northeastern flank (Fig. 4). In contrast, $F_{met}$ around A-23A was generally within background levels (1.19–2.32%), indicating minimal enhancement of glacial meltwater input.

Although $F_{met}$ derived from salinity and $\delta^{18}O$ integrates contributions from both glacial melt and precipitation, we attribute elevated values around A-76A to glacial iceberg melt given the strong spatial congruence with the iceberg position. Nonetheless, $F_{met}$ exhibited considerable heterogeneity, ranging from background levels up to 3.94% despite sampling locations being in relatively close proximity. Such fine-scale variability likely reflects the complex meltwater field produced by iceberg geometry and the location of discrete meltwater sources from iceberg fragments, as observed with giant iceberg A-68A[36]. This emphasises that surface sampling alone may underestimate or spatially blur patterns of glacial freshwater release.

These observations demonstrate that A-76A delivers measurable glacial meltwater to the surface ocean, establishing a physical pathway through which icebergs can influence ocean biogeochemistry. This meltwater may contribute to nutrient supply through direct entrainment of terrigenous material, providing micronutrients, through sidewall melting that entrains ambient seawater and drives vertical mixing, or through a combination of these processes. In contrast, the lack of an enhanced surface meltwater signal around A-23A suggests that iceberg melt is not exerting a strong influence on upper-ocean hydrography, likely restricting iceberg-induced nutrient delivery and thereby limiting both the initial stimulation and ongoing maintenance of a biological response during the sampling period.

**Macronutrient distributions differ between icebergs**

Macronutrient distributions and concentrations provide insight into whether iceberg passage supplies the nutrients required to sustain enhanced primary productivity, or whether background concentrations limit the biological response. Macronutrient concentrations were substantially higher around A-23A than A-76A (nitrate ($NO_3^-$): 26.6 vs. 14.6 µM; phosphate ($PO_4^{3-}$): 1.9 vs. 1.3 µM; silicic acid (DSi, $Si(OH)_4$): 76.5 vs. 61.3 µM, two-samples $t$-test, all $p < 0.001$). In contrast, nitrite ($NO_2^-$) was elevated around A-76A (0.13 vs. 0.08 µM; $p < 0.001$), though it contributed <1.8% of total $NO_x$ for both icebergs. Around A-76A, macronutrient distributions were more heterogeneous than A-23A, with consistently lower concentrations at the iceberg's northeastern flank (Fig. 5).

To place these concentrations in the context of the biological response observed around A-76A, we consider the macronutrient demand required to support the magnitude of the chl-$\alpha$ enhancement identified with both in situ and satellite observations. Using a representative Southern Ocean chl-$\alpha$ - to- carbon ratio of 0.03 g g$^{-1}$[37] and Redfield stoichiometry, the maximum

chl-$\alpha$ concentration from the in situ sensor (10.3 mg m$^{-3}$; Fig. 3) corresponds to a nitrate requirement of around 5.4 µM and a phosphate requirement of around 0.34 µM. Taking the maximum chl-$\alpha$ concentration from the satellite imagery (41 mg m$^{-3}$; Fig. 3), the nitrate requirement would be around 17.2 µM, and the phosphate requirement around 1.1 µM for standing biomass alone. While actual Southern Ocean phytoplankton stoichiometry can deviate from Redfield ratios, these calculations provide a useful first-order estimate of whether observed nutrient concentrations could support the observed chlorophyll concentrations.

Macronutrient demands estimated from the in situ chl-$\alpha$ maximum are below the range of nitrate and phosphate concentrations observed in surface waters around A-76A, whereas demands inferred from the satellite-derived chl-$\alpha$ maximum exceed concentrations at the lower end of the observed range. Regardless of these instantaneous comparisons, the persistence of elevated chl-$\alpha$ over the 8-day period represented by the satellite-derived composite (Fig. 3) implies sustained uptake through time rather than a single, transient pulse. Losses due to grazing, export, and physical dilution would therefore require continued macronutrient resupply to maintain a bloom of this magnitude. This supports the interpretation that iceberg-associated processes around A-76A acted to sustain, rather than merely initiate, enhanced productivity. In contrast, despite higher background macronutrient concentrations around A-23A, the absence of measurable meltwater input and lack of clear evidence of physical perturbation of the upper ocean suggest that the presence of the iceberg did not supply micronutrients previously limiting productivity, such as iron, meaning a comparable biological response was not triggered.

To investigate differences in nutrient supply and utilisation, we examined the stoichiometric tracers N* and Si*, which were calculated following commonly used formulations. Specifically, N* was calculated as $[NO_3^-]$ - 16$[PO_4^{3-}]$ (following ref. 38) and Si* as $\left[Si(OH)_4\right] - \left[NO_3^-\right]$ (following ref. 39). Mean N* was significantly higher around A-23A than A-76A (−3.5 vs −5.5 µM; $p = 0.012$), whereas mean Si* showed no significant difference between the two (49.9 vs. 46.7 µM; $p = 0.268$). Both tracers exhibited similar spatial patterns to macronutrient concentrations around A-76A, with minima on the northern flank (Fig. 6).

The wide range in N* around both icebergs indicates spatial heterogeneity in the coupling of nitrate and phosphate cycles, reflecting variation in both nutrient utilisation and supply. Consumption above the Redfield ratio lowers N*[40]; hence, the elevated $NO_3^-$ : $PO_4^{3-}$ ratios observed around both icebergs (Supplementary Note 1, Supplementary Fig. S3) suggest that non-diatom productivity could dominate locally[41]. Alternatively, N* may be reduced by upwelling of CDW, which typically exhibits lower N* than surface waters[42]. While there was no correlation of $F_{met}$ with phosphate or DSi concentrations (Supplementary Table 1), significant negative correlations were observed between $F_{met}$ and N* ($R^2 = 0.33$, $p = 0.026$, Fig. 7), $NO_3^-$ ($R^2 = 0.28$, $p = 0.042$), and $NO_2^-$ ($R^2 = 0.32$, $p = 0.029$) around A-76A (Fig. 7). This association between elevated $F_{met}$ and more negative N* is

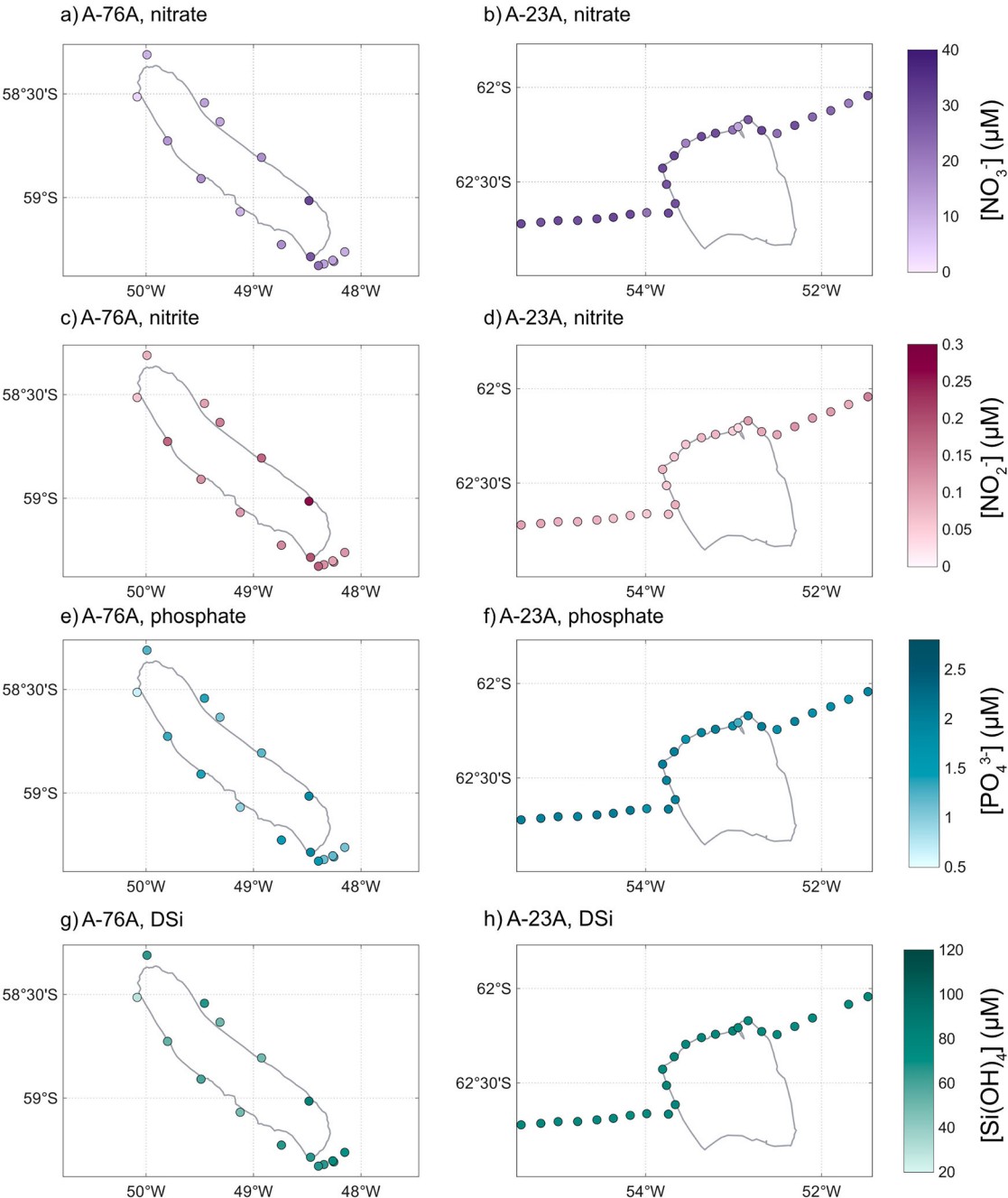

**Fig. 5 | Macronutrient concentrations surrounding giant icebergs A-76A and A-23A.** Panels show nitrate (**a, b**), nitrite (**c, d**), phosphate (**e, f**), and dissolved silicic acid (**g, h**) around A-76A (left) and A-23A (right), respectively. All concentrations are in μM.

consistent with iceberg-associated meltwater being linked to intermittent entrainment of CDW into the upper ocean.

Despite this physical supply, surface $NO_3^-$ concentrations decrease with increasing $F_{met}$, consistent with enhanced biological uptake in locally enriched regions. The observed nitrate drawdown, therefore, likely reflects the combined effects of CDW-derived macronutrient resupply and elevated phytoplankton consumption. Although $NO_2^-$ constitutes a minor fraction of the total $NO_x$ pool, its negative correlation with $F_{met}$ is consistent with these nitrate-driven dynamics and is unlikely to represent an independent control on nutrient cycling.

High Si* values around both icebergs indicate that DSi is replete relative to nitrate and does not limit diatom growth[39]. Around A-23A, Si* ranged from 44.7 to 62.6 μM, largely consistent with upwelled CDW[43]. Around A-76A, Si* ranged from 25.3 to 62.9 μM; the lower minimum may

reflect partial DSi drawdown from upwelled CDW, potentially occurring at Si:N ratios > 1:1 under iron limitation[43,44]. Nevertheless, the broad range of Si* in CDW (10–55 μM[43]) suggests that variability in source water composition may also contribute to the observed spatial gradients.

Together, macronutrient concentrations and stoichiometric tracers demonstrate that iceberg passage can fundamentally alter the balance between nutrient supply and biological utilisation in the surface ocean, with contrasting outcomes for A-76A and A-23A. Around A-76A, spatially variable macronutrient fields, negative N* values, and their association with elevated $F_{met}$ are consistent with iceberg-stimulated entrainment of CDW to the upper ocean, providing a mechanism for macronutrient resupply capable of sustaining high-magnitude phytoplankton biomass once growth has been initiated. In contrast, the more homogenous macronutrient distributions around A-23A, and the background level chl-$\alpha$ concentrations,

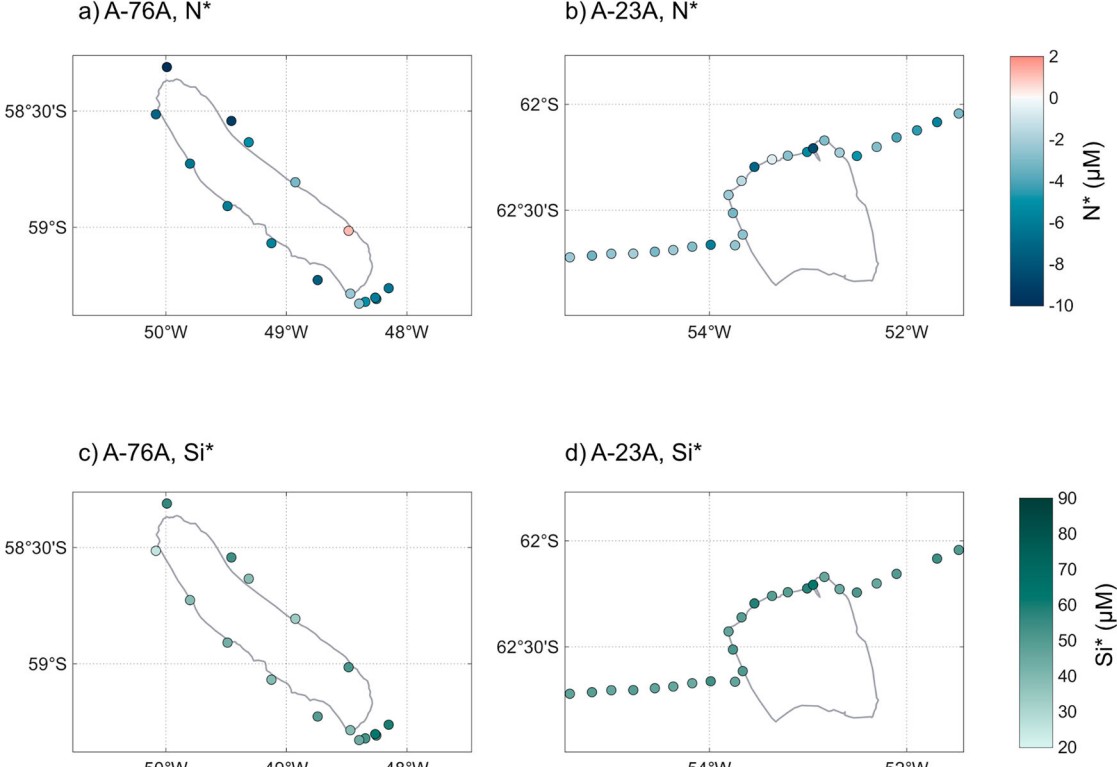

**Fig. 6 | Nutrient deviations surrounding giant icebergs A-76A and A-23A.** Panels (**a**) and (**b**) show N* (µM), while panels (**c**) and (**d**) show Si* (µM) around A-76A (left) and A-23A (right), respectively.

indicate that the absence of iceberg-driven physical perturbation precluded a biological response. However, macronutrient concentrations and stoichiometric tracers integrate the effects of both physical supply and biological uptake, leading to inherent ambiguity when considered alone. We therefore examine silicon isotopic composition as a complementary tracer to provide additional constraint on nutrient supply and utilisation in regions influenced by iceberg passage, and to further assess the role of macronutrient resupply in enabling the maintenance of elevated productivity around giant icebergs.

### Silicon isotopes reveal iceberg-driven nutrient utilisation

Silicon isotopes in dissolved silicic acid ($\delta^{30}Si_{DSi}$) reflect the integrated effects of DSi supply and diatom utilisation, as biological uptake preferentially incorporates lighter isotopes, leaving the residual pool isotopically heavier[45]. Subsequent resupply with isotopically lighter DSi, from upwelled CDW, resets the signal and enables continued utilisation. Here, we use $\delta^{30}Si_{DSi}$ to assess how iceberg-driven physical perturbations regulate macronutrient resupply and, in turn, shape biological uptake around the giant icebergs A-76A and A-23A.

Surface $\delta^{30}Si_{DSi}$ differed between the two icebergs (Fig. 8). Around A-76A, $\delta^{30}Si_{DSi}$ ranged from 1.26 to 2.19‰, significantly heavier than both A-23A (0.83–1.18‰; two-samples $t$-test; $p < 0.001$) and the source CDW (0.96 ± 0.02‰; $p < 0.001$). In contrast, $\delta^{30}Si_{DSi}$ around A-23A was indistinguishable from CDW ($p = 0.099$). Spatial variability was pronounced around A-76A, with heavier values to the northeast, whereas A-23A showed little heterogeneity. $\delta^{30}Si_{DSi}$ and DSi were strongly negatively correlated around A-76A ($R^2 = 0.93$, $p < 0.001$) but not around A-23A ($p = 0.24$).

At A-23A $\delta^{30}Si_{DSi}$ values indistinguishable from the CDW source indicate minimal isotopic fractionation and suggest that upwelled DSi was not substantially modified by biological uptake. This interpretation aligns with high ambient DSi concentrations and the absence of a $\delta^{30}Si_{DSi}$ - DSi relationship, consistent with relatively homogenous mixing rather than active utilisation. The slightly lighter minimum (0.83 ± 0.10‰) compared

with CDW may reflect minor inputs from biogenic[46] or lithogenic[47–49] silica dissolution, though this difference is only slightly greater than analytical uncertainty.

In contrast, $\delta^{30}Si_{DSi}$ around A-76A is both elevated and highly variable (Fig. 9), reflecting substantial isotopic fractionation driven by diatom uptake. The observed range exceeds that reported for single blooms or seasonal cycles in previous Southern Ocean studies[50–53], indicating intense and spatially heterogeneous fractionation. The steep negative relationship between $\delta^{30}Si_{DSi}$ and DSi is consistent with progressive DSi drawdown—neither parameter varies systematically with latitude ($p = 0.52$ and $p = 0.064$), suggesting variability is not driven by the large-scale south-north gradient observed in other studies[54–56]. The spatial heterogeneity may reflect a combination of local nutrient resupply, differential bloom stages, or mixing of upwelled CDW with surface waters, with lighter DSi inputs to some areas, and heavier residual $\delta^{30}Si_{DSi}$ in others. These patterns, together with elevated chl-α, indicate a dynamic regime of simultaneous DSi consumption and replenishment, consistent with upwelling sustaining active diatom productivity rather than iceberg passage triggering only a transient bloom response.

To evaluate whether the observed fractionation reflected a single DSi input or continuous nutrient resupply, simple mass balance calculations and Rayleigh fractionation models were fitted to the $\delta^{30}Si_{DSi}$ - DSi data (see Methods). The data were reproduced well with both open ($R^2 = 0.93$) and closed ($R^2 = 0.86$) systems, but the open system provided the best fit (RMSE = 0.07, Fig. 4) with an enrichment factor ($\epsilon = -1.76$‰) within the range reported for previous Southern Ocean water column studies[51–53,55–59] and that reported for Southern Ocean diatom assemblages[60]. This outcome supports ongoing nutrient resupply through upwelling, allowing elevated chl-α concentrations to be maintained, rather than a single input bloom.

Variability in $\delta^{30}Si_{DSi}$ around A-76A was also coupled to macronutrient dynamics. $\delta^{30}Si_{DSi}$ correlated negatively with phosphate ($R^2 = 0.72$, $p = 0.001$) and nitrate ($R^2 = 0.56$, $p = 0.008$, Fig. 8), indicating that diatom productivity influenced the drawdown of both DSi and other

**Fig. 7 | Relationship between meteoric water fraction, N\*, $NO_3^-$, and $NO_2^-$ around giant icebergs A-76A and A-23A.** The relationship between $F_{met}$ and N\* (**a**), $NO_3^-$ (**b**), and $NO_2^-$ (**c**) is shown for waters surrounding icebergs A-76A (green) and A-23A (orange). Linear regressions are fitted for each iceberg, with corresponding equations, coefficients of determination ($R^2$), and $p$-values.

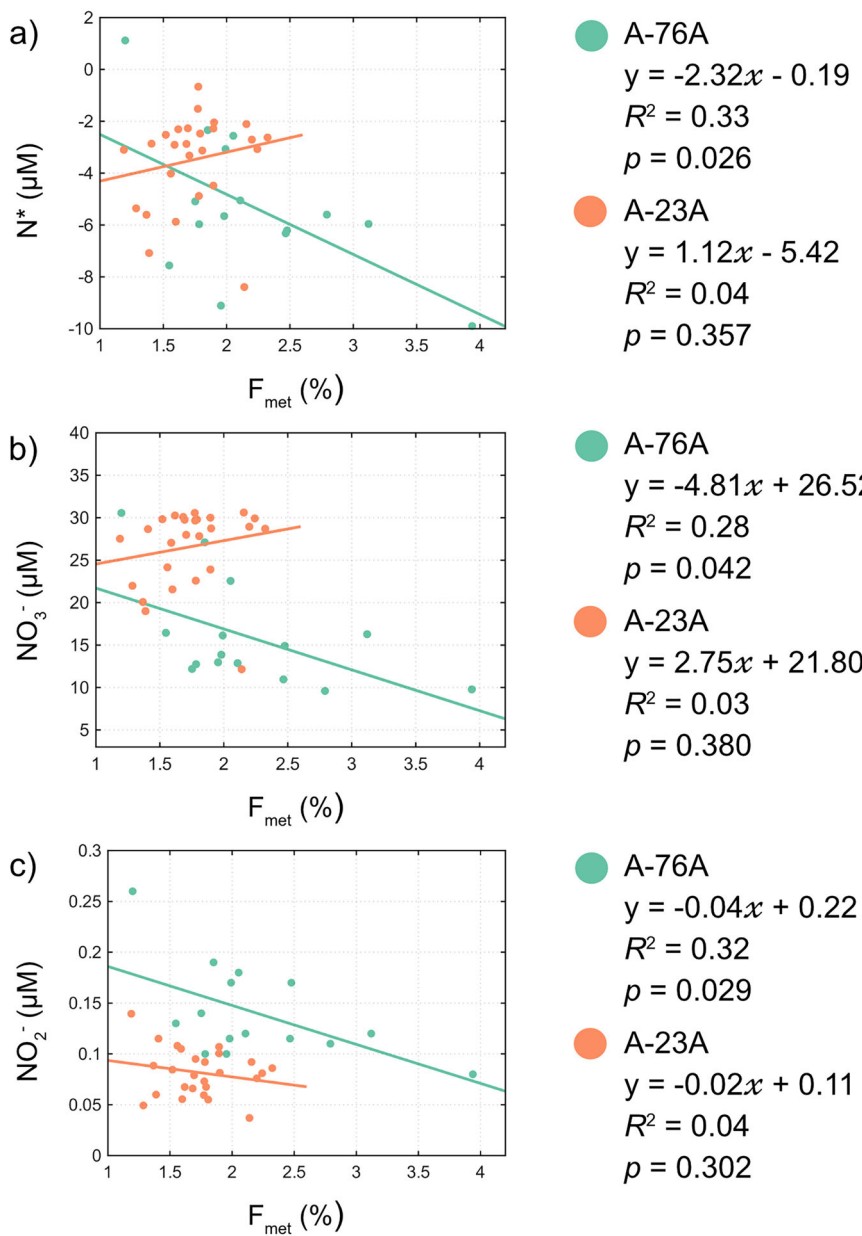

macronutrients. These correlations further support the interpretation that biological uptake, enabled by sustained macronutrient resupply, exerted strong biological control on surface nutrient distributions around A-76A.

Together, these contrasts highlight the divergent regimes of silica cycling around the two icebergs. A-23A represents a system dominated by recent CDW upwelling with little biological modification, whereas A-76A reflects an actively fractionating, diatom-dominated environment with ongoing nutrient resupply. The silicon isotope signatures therefore record the imprint of iceberg-driven macronutrient resupply from upwelling that enables continued biological utilisation of DSi once growth has been stimulated, supporting sustained productivity enhancement in surface waters. When considered alongside hydrographic, meltwater, macronutrient, and chl-$\alpha$ observations, these results support a mechanistic framework in which micronutrient inputs from iceberg meltwater stimulate initial bloom development, while continued macronutrient resupply from CDW upwelling is required to maintain high biomass accumulation. In this context, $\delta^{30}Si_{DSi}$ provides confirmation that iceberg-ocean interactions play a critical role in sustaining productivity enhancement, rather than merely triggering short-lived responses.

## Conclusions

Our findings show that giant icebergs influence surface ocean biogeochemistry in contrasting ways, shaped by differences in melt-driven freshwater input, nutrient entrainment, and background ocean structure. Around A-76A, meteoric water, macronutrient patterns, and silicon isotopes together indicate a dual control mechanism: micronutrient input from meltwater initiates bloom formation, while subsequent nutrient resupply via iceberg-driven upwelling enables the sustained expression of elevated productivity. In contrast, A-23A exhibits limited meltwater influence, more homogeneous macronutrient fields, and minimal biological uptake, reflecting a system in which iceberg passage neither initiates nor sustains an obvious biological response. These results highlight that physical modulation of macronutrient supply plays a key role in maintaining iceberg-driven productivity enhancement, but must be considered alongside direct micronutrient inputs, iceberg history, and environmental preconditioning. As iceberg discharge intensifies with continued Antarctic ice sheet retreat, incorporating this initiation-maintenance dual-control framework into predictive models will be critical for constraining the feedbacks between cryospheric change, ocean biogeochemistry, and the global carbon cycle.

**Fig. 8 | Relationships between macronutrient concentration and silicon isotope composition and in waters surrounding giant icebergs A-76A and A-23A.** Panel **a** shows the relationship between DSi concentration concentration ($\mu M$) and $\delta^{30}Si_{DSi}$ (‰) for waters surrounding icebergs A-76A (green) and A-23A (orange), and initial CDW waters (black) (**a**). Panels (**b–d**) show the relationship between $\delta^{30}Si_{DSi}$ and DSi (**b**), nitrate (**c**), and phosphate (**d**). Linear regressions are fitted for each iceberg, with corresponding equations, coefficients of determination ($R^2$), and $p$-values. Error bars represent ±0.100 ‰, equivalent to twice the maximum standard deviation of the reference standards.

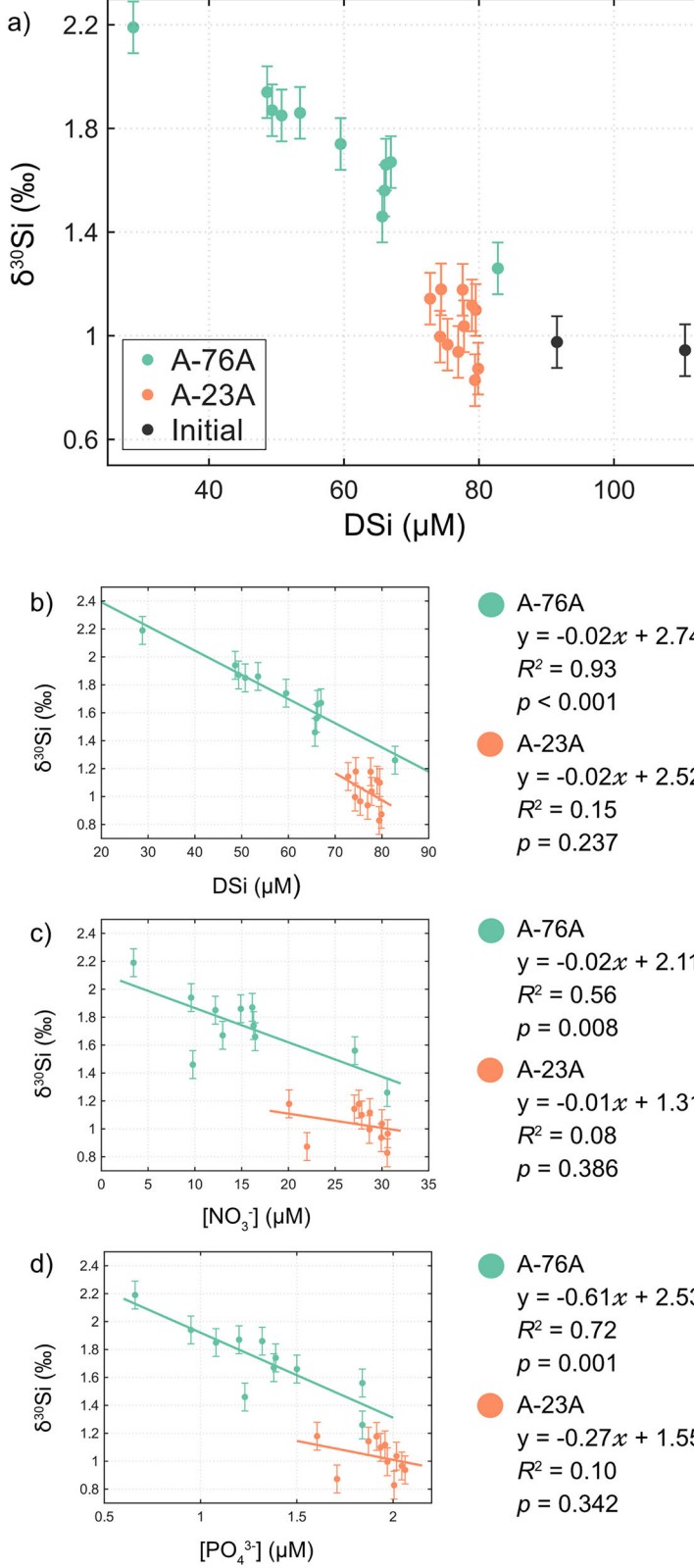

## Methods

### Sample and collection

Seawater samples for biogeochemical analysis were opportunistically collected around giant icebergs A-76A and A-23A during the cruises DY158 and SD033, respectively.

The RRS Discovery (DY158) encountered iceberg A-76A (area: ~3200 km²) on 25th January 2023, northwest of the South Orkney Islands (Figure 1). The vessel completed a circumnavigation of the iceberg at a distance of ~400 m, with sampling from the surface uncontaminated seawater supply at a 20 min maximum frequency

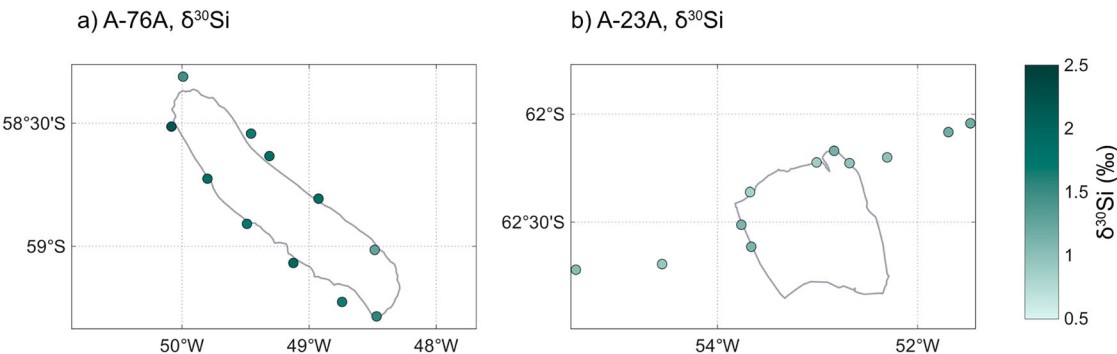

**Fig. 9 | Dissolved silicon isotope composition surrounding giant icebergs A-76A and A-23A.** Panels show $\delta^{30}Si_{DSi}$ (‰) in the vicinity of icebergs A-76A (**a**) and A-23A (**b**).

while maintaining an approximately constant speed, yielding broadly evenly spaced stations.

The RRS Sir David Attenborough (SD033) encountered iceberg A-23A (area: ~3900 km²) on 1st December 2023 close to Powell Basin (Fig. 1). Sampling commenced ~75 km from the iceberg's western side, proceeded along the northern side at a distance of ~400 m, and continued to a position ~75 km to the northeast. Samples for different analyses were collected at either 15 or 30 min intervals while the ship travelled at approximately constant speed, producing an evenly spaced transect.

Samples for dissolved nitrate, nitrite, phosphate, and dissolved silica (DSi) analyses, as well as oxygen isotopes, were collected from the research vessels' uncontaminated seawater systems. Nutrient samples (nitrate, nitrite, phosphate) and DSi samples were filtered through an in-line Cytiva AcroPak filter (0.8/0.45 μm) into acid cleaned (10% reagent grade HCl) 60 ml and 250 ml HDPE bottles, respectively. Nutrient samples were immediately frozen at −20 °C and DSi samples were stored in the dark at 4 °C until subsequent analysis on land. Seawater oxygen isotope samples were collected unfiltered into 50 ml glass bottles, sealed with rubber stoppers and aluminium crimp seals, and transported in the dark at 4 °C.

### Sensor data collection
Underway sensor measurements provide continuous, high-resolution observations of surface ocean properties along the ship's track, complementing discrete sampling by capturing the conditions present during sample collection. In this study, temperature, salinity, and chl-α were measured from the uncontaminated seawater systems on both research vessels.

The uncontaminated seawater system inlet was at 5 m depth on the RRS Discovery and 7 m on the RRS Sir David Attenborough. Underway temperature and salinity measurements were taken with a Seabird Scientific SBE 45 thermosalinograph with a remote SBE 38 temperature sensor located near the water inlet of the vessels. Salinity was calibrated against in situ samples collected at 3-hourly intervals on DY158 and 4-hourly intervals on SD033, and analysed on a Guildline Autosal 8400B salinometer against IAPSO standard seawater batch P164 (DY158) and P165/P167 (SD033). Salinities were measured on the Practical Salinity Scale.

Fluorescence data were collected using a WS3S WETStar fluorometer and were converted to chl-α using factory calibrations and were not further calibrated against in situ data.

After removing spikes in the data streams, underway data were averaged at 5-minute intervals.

### Oxygen isotopes in seawater
Samples were analysed for their oxygen isotope composition, $\delta^{18}O$, the standardised ratio of $H_2^{18}O$ to $H_2^{16}O$ in seawater, using the $CO_2$ equilibration method with an Isoprime 100 mass spectrometer plus Aquaprep device. Measurements were calibrated against international primary standard VSMOW2. The long-term mean of all seawater replicates is better than 0.04‰ (1σ).

Quantitative information on freshwater provenance is obtained using a three-endmember mass balance equation, assuming each sample is composed of three components: sea ice melt, meteoric water, and an ambient saline oceanic endmember of CDW. Separate contributions are determined via

$$F_{sim} + F_{met} + F_{cdw} = 1 \tag{1}$$

$$S_{sim} \cdot F_{sim} + S_{met} \cdot F_{met} + S_{cdw} \cdot F_{cdw} = S \tag{2}$$

$$\delta_{sim} \cdot F_{sim} + \delta_{met} \cdot F_{met} + \delta_{cdw} \cdot F_{cdw} = \delta \tag{3}$$

where:

S and $\delta$ are the salinity and $\delta^{18}O$ of a water sample;
$F_{sim}$, $F_{met}$, and $F_{cdw}$ are the fractions of sea ice melt, meteoric water, and CDW;
$S_{sim}$, $S_{met}$, and $S_{cdw}$ are the salinities of the pure endmembers; and
$\delta_{sim}$, $\delta_{met}$, and $\delta_{cdw}$ are the corresponding $\delta^{18}O$ of the endmembers.

Determining realistic freshwater contributions requires accurate endmember choices. Oceanic and sea ice endmembers in the region are well established, as described in Meredith et al.[61]. Meteoric endmembers for this study were chosen to be those given by Brown et al.[62] to account for the icebergs' origins from the Filchner and Ronne ice shelves in the Weddell Sea. Endmembers used are described in Supplementary Table S2.

The largest uncertainty propagated in the freshwater fraction determination is from the mean meteoric water $\delta^{18}O$ endmember as it represents a combination of local seasonally variable precipitation and glacial melt. Sensitivity studies have found that uncertainties in the final freshwater fractions are better than 1% of the total sampled fluid volume[61,62].

### Macronutrient analysis
Nitrate, nitrite, and phosphate from around iceberg A-76A were analysed on shore at Plymouth Marine Laboratory (PML), and from around A-23A at the National Oceanography Centre (NOC). DSi samples from both icebergs were analysed at the British Antarctic Survey (BAS).

At PML and NOC, samples for nitrate, nitrite, and phosphate were thawed using international GO-SHIP protocols[63]. Following the analytical protocols of Woodward and Rees[64], samples were analysed on SEAL Analytical AAIII segmented flow colorimetric autoanalysers at PML and BAS, and a SEAL QuAAtro 39 at NOC. Sample handling and manipulation followed Becker et al.[63] as closely as possible. Calibration standards were prepared with low-nutrient seawater and the analytical results were quality controlled by reference to analysed certified reference materials (CRMs) supplied from Kanso Ltd. (Japan). Raw data were further corrected to ambient ocean salinity and pH.

## Dissolved silicon isotopes

Samples were acidified with 0.2% v/v 1M HCl (reagent grade, distilled 'in house') in the Department of Earth Sciences laboratories, University of Cambridge, to dissolve any silicon precipitates that may have formed during storage and to achieve a pH between 5–7 for subsequent processing. A 2.5 ml aliquot was removed for DSi concentration analysis described above. DSi was precipitated from seawater using the 'Magnesium Induced Co-precipitation' (MAGIC) method, a two-step quantitative $Si(OH)_4$ scavenging by brucite $(Mg(OH)_2)$[58,65,66]. Samples were purified using AG50W-X12 cation exchange resin, adapted from Georg et al.[67]. The AG50W-X12 resin was pre-cleaned and rinsed with acid (3M, 6M, and concentrated HCl) and Milli-Q ($18.2\,M\Omega \cdot cm$) before samples and standards were passed through the columns and eluted with Milli-Q. Si concentrations were checked, confirming all Si yields from the MAGIC method were above 90%.

Silicon isotopic compositions were determined on a Thermo Scientific Neptune multicollector inductively coupled plasma mass spectrometer (MC-ICP-MS) at the University of Cambridge Department of Earth Sciences. Silicon isotopic compositions ($\delta^{30}Si_{DSi}$) were calculated as the per mil deviation from the primary reference standard NBS28 (NIST RM 8546) measured immediately before and after each sample, and expressed as

$$\delta^{30}Si(‰) = \left[\frac{^{30}Si_{sample} - \ ^{30}Si_{standard}}{^{30}Si_{sample} - \ ^{30}Si_{standard}}\right] \times 1000 \tag{4}$$

0.1 ppm of an ICP-MS magnesium (Mg) standard was added to the samples to correct for instrumental mass bias by estimating the isotopic fractionation occurring within the machine during measurement through monitoring the $^{26}Mg/^{25}Mg$ ratio[68,69]. Sulphuric acid was added to account for differences in residual sulphate in solutions being analysed. Instrument blank levels were measured at less than 0.8% of the main signal and were subtracted from each sample and standard analysis. Procedural blanks were indistinguishable from instrument blanks.

Most samples were measured in duplicate, with a maximum standard deviation of ±0.092‰. Secondary reference materials (CRMs) were analysed between every 5 samples throughout the analytical procedure to assess average precision and long-term reproducibility. These included: ALOHA 1000 m seawater (a deep Pacific Ocean seawater standard; 1.23 ±0.05‰ ($1\sigma$), $n = 12$[70]), diatomite (a siliceous sediment standard; 1.31 ±0.03‰, $n = 5$), and LMG (a siliceous sponge spicule standard; (−3.31 ±0.04‰, $n = 7$), which all agree with published values[70–72]. Errors are reported as twice the maximum standard deviation of the reference standards throughout the session (±0.10), which all sample standard deviations are below.

To evaluate the processes governing $\delta^{30}Si_{DSi}$ variability and to distinguish between closed- and open-system behaviour, simple mass balance calculations were applied following established approaches for marine silicon fractionation e.g., refs. [51,55,57]. These models describe the evolution of $\delta^{30}Si_{DSi}$ as diatom uptake fractionates silicon between the dissolved and biogenic silica pools.

Two conceptual endmembers were considered. In the closed system, a single DSi input is supplied prior to biological utilisation, and no further resupply occurs. Fractionation proceeds as the DSi reservoir is progressively consumed, following a Rayleigh-type model:

$$\delta^{30}Si_{DSi} = \delta^{30}Si_{DSi\ initial} + \varepsilon \ln(f) \tag{5}$$

where $f$ is the fraction of DSi remaining ($f = [DSi]/[DSi]_{initial}$), $\varepsilon$ is the isotopic enrichment factor (in‰), and $\delta^{30}Si_{DSi\ initial}$ represents the isotopic composition of the initial DSi source.

In contrast, the open system represents a steady-state regime in which DSi is continually resupplied to surface waters while being consumed by diatom uptake. Under this assumption, the isotopic composition of the residual DSi pool is expressed as:

$$\delta^{30}Si_{DSi} = \delta^{30}Si_{DSi\ initial} + \epsilon(1 - f) \tag{6}$$

where parameters are defined as above.

For both models, $\varepsilon$ and $\delta^{30}Si_{DSi\ initial}$ were determined by fitting the equations to the observed $\delta^{30}Si_{DSi}$ - DSi relationships. The initial DSi concentration, $[DSi]_{initial}$ was defined from measured CDW concentrations on each cruise, following the methods above.

## Data availability

Observational data used in this study are available from the UK Polar Data Centre (https://doi.org/10.5285/89b12c53-1244-4fbf-8699-262fa4c97b27). Previously published datasets are cited in the text and figure captions.

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

## Acknowledgements

The authors are grateful for the support of the officers, crew, and scientists aboard the DY158 and SD033 cruises for all assistance and support. We thank, in particular, Dr Ryan Saunders, Chief Scientist of DY158, for allowing time for the transit around iceberg A-76A. We also thank Dr Rhiannon Jones for assistance with laboratory analyses. This study was funded by the Natural Environment Research Council via C-CLEAR Doctoral Training Partnership (NE/S007164/1) (L.R.T.), BIOPOLE (NE/W004933/1) (K.R.H., R.N.C.S., M.P.M., A.M., E.M., C.A., M.J.L, and C.M.), and SiCLING (NE/X014819/1) (H.P., K.R.H., and H.M.W.).

## Author contributions

L.R.T.: conceptualisation, project administration, fieldwork, dissolved silicon isotope analysis, nutrients analysis, data analysis, writing- original draft, writing- review and editing; H.P.: dissolved silicon isotope analysis, supervision, writing- review and editing; K.R.H.: conceptualisation, supervision, resources, data analysis, writing- review and editing; R.N.C.S.: data analysis, writing- review and editing; M.P.M.: fieldwork, writing- review and editing; A.M.: fieldwork, writing- review and editing; E.M.: nutrients analysis, writing- review and editing; E.M.S.W.: nutrients analysis, writing-review and editing; C.A.: oxygen isotope analysis, writing- review and editing; M.J.L.: oxygen isotope analysis, writing- review and editing; E.P.A.: fieldwork, data analysis; H.M.W.: resources, supervision, writing- review and editing; C.M.: conceptualisation, supervision, resources, fieldwork, data analysis, writing- review and editing.

## Competing interests

The authors declare no competing interests.
