## [Transparent Peer Review file · Communications Earth & Environment]

Giant icebergs impact regional biogeochemical cycling in the Southern Ocean

Corresponding Author: Ms Laura Taylor

Version 0:

Decision Letter:

Dear Ms Taylor,

Your manuscript titled "Giant iceberg behaviour impacts regional biogeochemical cycling in the Southern Ocean" has now been seen by 3 reviewers, whose comments are appended below. You will see that they find your work of some potential interest. However, they have raised substantial concerns that must be addressed. In light of these comments, extensive revisions will be required before we can further consider the manuscript for publication. We would, however, be interested in considering a revised version that fully addresses these serious concerns.

In particular, please ensure that the revised manuscript meets the following editorial thresholds:

1. Present a compelling case that your interpretations and conclusions regarding the mechanisms explaining the differences in productivity responses observed between the two analysed icebergs are robust and fully supported by your analyses.
2. Clearly articulate the implications of your findings for ocean biogeochemistry.

We hope you will find the reviewers' comments useful as you decide how to proceed. If additional work allows you to either incorporate or refute these criticisms, we will be happy to look at a substantially revised manuscript. If you choose to take up this option, please either highlight all changes in the manuscript text file, or provide a list of the changes to the manuscript with your responses to the reviewers.

When resubmitting, please provide a point-by-point response to the reviewers' comments. Please submit your responses as a separate file, distinct from your cover letter where you can add responses to the Editors' comments that you do not want to be made available to the reviewers. Word files are preferred. We recommend that any figures, tables or graphs that are included in the response to reviewers are also included in the main article or Supplementary Information.

If the revision process takes significantly longer than three months, we will be happy to reconsider your paper at a later date, as long as nothing similar has been accepted for publication at Communications Earth & Environment or published elsewhere in the meantime.

Please use the following link to submit your revised manuscript, point-by-point response to the reviewers' comments with a list of your changes to the manuscript text (which should be in a separate document to any cover letter), a tracked-changes version of the manuscript (as a PDF file) and any completed checklist:

Link Redacted

Please do not hesitate to contact us if you have any questions or would like to discuss the required revisions further. Thank you for the opportunity to review your work.

Best regards,

Jose Luis Iriarte Machuca, PhD
Editorial Board Member
Communications Earth & Environment

Nicola Colombo, PhD
Associate Editor, Communications Earth & Environment
Consulting Editor, Communications Sustainability

EDITORIAL POLICIES AND FORMAT

If you decide to resubmit your paper, please ensure that your manuscript complies with our editorial policies and complete and upload the checklist below as a Related Manuscript file type with the revised article:

- Behavioural and social science
- Ecological, evolutionary & environmental sciences
- Life sciences

For your information, you can find some guidance regarding format requirements summarized on the following checklist: (<https://www.nature.com/documents/commsj-phys-style-formatting-checklist-article.pdf>) and formatting guide (<https://www.nature.com/documents/commsj-phys-style-formatting-guide-accept.pdf>).

REVIEWER COMMENTS:

Reviewer #1 (Remarks to the Author):

Review of 15978

This paper examines nutrient uptake around two giant icebergs in the Weddell Sea in 2023. It finds very different behaviour in terms of many parameters, and particularly silicon and oxygen isotope fractionation, with the more recently calved A76A showing much more dynamic uptake than the older A23A. The paper is generally well written and clearly explained. The text and methods are well separated on the whole, and the supplementary figures are appropriate. I do not have major concerns about the paper and believe it is worthy of publication after a few points are expanded upon. These are explained below.

Point 1: While the paper clearly discusses the contrasting origins and life histories of the two icebergs I felt more might have been made of the decades difference in exposure to the ocean felt by the contrasting bergs. A23A had much more time to be changed by the ocean while it was grounded for decades, compared to the short ocean lifetime of A76A. How much might the difference in nutrient behaviour be explained by this exposure? For example, if sediments contained in the ice are more likely to be near the edges, much more of these would have been lost by A23A before it entered the ACC than in A76A's case. Is it possible to estimate this size change from satellite images, so as to speculate on this effect?

Point 2: The origin and trajectory of the two icebergs probably contains some indication of each berg's depth. This is not discussed, but may be important for the upwelling tendency and behaviour exhibited during the cruises. The underwater shape may also be rather different, given the different ocean exposure times too. Some discussion of the impacts of these parameters on the nutrient uptake would be good.

Point 3: Section 2.3: I could not find formal definitions of Fmet and Fsim. This should be presented at first appearance.

Point 4: section 3.6 Data availability This needs to be added. It currently just says "TBC"

Reviewer #2 (Remarks to the Author):

Review of Research Article MS# COMMSENV-25-5511-T

Giant iceberg behaviour impacts regional biogeochemical cycling in the Southern Ocean L.R. Taylor et al.

The manuscript titled "Giant iceberg behavior impacts regional biogeochemical cycling in the Southern Ocean" delves into the influence of giant icebergs on biogeochemical cycling by altering local circulation and nutrient availability. Conducted in 2023, the study involved two British research vessels sampling around two large icebergs, A-76A and A-23A. These icebergs exhibited contrasting conditions, with one supporting meltwater-induced diatom blooms while the other showed no visible biological changes due to meltwater. The authors employed silicon isotope fractionation as a tracer for diatom

biological uptake. They emphasize that the contrasting icebergs they measured illustrate the highly heterogeneous effects of giant iceberg transport and melt on local biogeochemistry, which is a combination of complex meltwater delivery and local circulation.

This study provides valuable measurements that illustrate the diverse effects of giant icebergs. As the authors rightly point out, the delivery and consequently the impact of giant icebergs will increase in the future.

Taylor et al. present as comprehensive a dataset as possible from opportunistic sampling and provide a valuable discussion that evaluates the diverse impacts of icebergs.

The manuscript is well-written, presenting observational results in an engaging manner to explore the impacts of giant icebergs on local biogeochemical cycling in the Southern Ocean. The data have been thoroughly analyzed, including a co-analysis of nutrients, including silicon and its isotopes. The authors present the data in visually appealing, albeit unclear, figures. With some modifications, I believe this paper will be a valuable contribution to the field.

The choice of the journal is appropriate, and this article is recommended for publication. However, there are a few minor and some moderate modifications that should be made. The authors may find the general and specific comments, questions, and linguistic corrections listed below helpful.

General

1. Figure clarity: Generally, the figures are aesthetically pleasing, but there are significant issues with their interpretability. Specifically, the color scale bars often don't match the spectrum of colors depicted in the figures. This makes it challenging to accurately interpret the range and values of the samples in hydrographic or biogeochemical variables.

a. Figure 1: Can the outlines be in different colors to distinguish between icebergs, Antarctic/island outlines, and other features? Additionally, the iceberg track appears to cross South Georgia. How did it do that?

b. Figure 2: It's difficult to see the variation in color in panel (b). Moreover, it seems that the colors shown in at least panel (a) and panel (d) aren't represented on the color scales. Panel (a) has more red colors, while panel (d) has quite green/yellow colors that aren't shown on the temperature and salinity color gradients. This makes it challenging to accurately interpret the figure.

c. Figure 4: Again, the colors in the figures are confusing. Why are there brown colors in panel (d)? Also, the colors in panel (a) are difficult to match to a percentage in Fmet. There are some medium brown colors that don't appear on the scale when there's an abrupt shift from dark brown/red to light grey/pink.

d. Figure 5: Once again, the colors in the figure don't match the scale bars. For example, silicic acid shows much darker blues than on the scale bar. Darker nitrite colors aren't shown on the scale bar of the light orange gradient. And the medium colors in nitrate plots aren't shown on the scale bar of dark red vs light grey/peach.

e. Figure 6: Colors again.

2. Framing of silicon isotope fractionation: The authors' abstract and conclusion assert that they demonstrate silicon isotopes are sensitive tracers of biological response. However, the use of silicon isotope fractionation to trace diatom growth is well-established, meaning the authors could be more modest in framing the novelty of their study. The paper's strengths lie in providing new observations of heterogeneous conditions around giant icebergs rather than introducing a novel technique.

3. Discussion of varying conditions: The abstract states: "These contrasting regimes reveal that the influence of icebergs on ocean biogeochemistry is highly heterogeneous, likely reflecting differences in meltwater delivery and local circulation." However, differences are not limited to local circulation. Not mentioned is the differing background conditions such as background levels of macronutrients. Blooms are supported when a limiting nutrient is supplied. Iceberg influence then only causes blooms when nutrients that were limiting are supplied. If macronutrients were sufficient beforehand, adding more, for instance due to meltwater-driven upwelling, won't stimulate further blooms. The differing background concentrations were discussed briefly. More discussion could be added to the manuscript to include not only varying iceberg delivery but also varying background conditions.

4. Separating Freshwater Fractions: In lines 174-176, the authors describe sea-ice formation. Negative values of Fsim indeed indicate sea ice formation, but the most negative values of Fsim occur where Fmet is high. This suggests that it's not real sea-ice formation but rather non-realistic endmembers causing the mixing model to produce positive or negative values when summed to 1. Furthermore, the authors mention sea ice melt and formation but then don't discuss them again. I agree that the majority of freshwater in the region is likely melt from the giant iceberg. However, perhaps more realistic endmembers could be chosen to better represent the system. If not, at least qualify what may be realistic.

5. Clarification of Si methodology: The methods are generally consistent with standard $\delta^{30}\text{Si}$ protocols. However, it would be helpful if the authors provided additional detail on the potential effects of the initial acidification step on dissolved Si integrity and recovery or confirmed that acidification did not introduce measurable Si loss prior to MAGIC processing.

Specific comments:

- Lines 10-11: 10-20% of the SO carbon flux.
- Lines 121-122: Are the icebergs referenced backwards? It seems colder near A76-A not higher.
- Line 135: Clarify that it is in discrete samples.
- Line 170: Reference supplemental figures in order. Not S3 first.
- Additionally, S1 and S2 are not referenced in the manuscript text? Why are they then included?
- Lines 189-196: There is a very abrupt transition from discussing meltwater to biogeochemistry. This paragraph could use more introduction and explanation. Additionally, the relationship is discussed but not presented as a figure.
- Line 191: N* needs to be introduced.
- Line 206-207: In the figure it looks like the opposite where A-23A had higher N*. Though it is difficult to tell due to the colors. (see figure comments above)
- Line 213: Reorder supplemental materials
- Line 223: "...the range observed around A-23A". Could the authors provide more context. The range is not stated and the colors do not show the full spectrum so the full range cannot be assessed.
- Line 276-279: Could be shown in figures? Might be interesting.

Reviewer #3 (Remarks to the Author):

I have now reviewed the manuscript entitled "Giant iceberg behaviour impacts regional biogeochemical cycling in the Southern Ocean" by Taylor and co-authors. This study presents interesting physical, chemical and nutrient data around 2 icebergs and endeavours to better constrain the mechanisms and factors whereby icebergs can enhance (or not) biological productivity in the Southern Ocean (SO).

Unfortunately, I find the paper to be unconvincing on several grounds (which I will detail below) and overall, I believe it lacks the scope and wide-ranging implications that are usually expected of publications in Nature Communications.

1) While I agree with the "first-degree" interpretations of the data I find that the authors do not in fact provide new, clear or well-justified mechanisms to explain the difference in productivity responses seen between the two icebergs, which they announce they set off to do in the introduction. Or if they do, such mechanisms are not clearly explained in the text and hardly discussed at all in the later part of the manuscript especially in the context of the different geographical locations and background conditions of the icebergs at the time of sampling, leaving it to the reader to infer what the factors that trigger these different responses actually are: is it melting that induces small scale upwelling of nutrients? The authors seem to point us in that direction. If yes, can the amount of melting be (semi)quantitatively linked to the amount of surplus productivity vs. background? Or is the main factor background temperatures which induces melting? Or is it background nutrient concentrations?... For example, it would have been very pertinent to show/measure background nutrient concentrations in the surface waters and compare it to the nutrient concentrations immediately around Iceberg A76 in particular (as this iceberg appears to impact productivity) in order to further ascertain whether there was indeed increased nutrient drawdown in the vicinity of this iceberg vs. in iceberg free zones. the background d30Si seem to corroborate this.

As it stands, the main conclusion of the paper, although not expressly stated, appears to be: higher SSTs = more melting = more influence of the iceberg on the surface/subsurface water = more productivity. This is hardly new or unexpected.

It is clear from the data that around A76 (compared to A23) background temperatures are higher, making the melting more intense, the background nutrient concentration is lower while background phytoplankton presence is higher, resulting in the drawdown of Si and increased productivity in patches trailing off the iceberg. Yet it is also clear that A76, at the time of sampling was in the ACC / fronts region while A23 was located southward in a hydrologically and dynamically very different setting. Could it be that the differences seen between the productivity responses around A76 and A23 are simply or at least partly due to the background conditions in the 2 locations as opposed to iceberg diversity* itself. What about stratification differences between both areas? What about background nutrient levels in the subsurface and surface? The potential influence of the oceanographic settings for the 2 icebergs on the productivity response is never properly discussed. Maybe it is irrelevant, but it should be mentioned/discussed.

*by the way, the alleged intrinsic "diversity" of both icebergs -as per the abstract- is never really established by the authors in the text (how are they different?). In what way do these differences impact the productivity response? This is not explained.

2) There are strong indications that some icebergs can deliver Iron to the surface ocean as they melt. (Geibert, Assmy et al. 2010) published about this mechanism specifically in the Weddell Sea. Today the SO is an HNLC region where productivity is vastly limited by the lack of Fe. While iron is vaguely mentioned in the text, the possibility of its influence on the productivity responses is not explored. This to me is an important omission. Iron limitation is inferred from N* by the authors but the rationale behind this is not well justified. Are there Iron data available to the authors? Could the icebergs, coming from different areas and calving at different time have a different iron load?

3) I find the claim (introduction) that d30Si offers a unique tool to look at the variable productivity responses to icebergs to be overblown as it would not show anything different from d15N for instance. If the authors had used d30Si in conjunction with d15N for instance, the iron limitation status in the surface water could have been established/inferred and this would have been a more powerful diagnostic tool.

4) The closing paragraphs remain vague and do not offer the clear mechanisms promised in the introduction. The implications of the findings (which are unclear) are not well-discussed.

Minor comments:

Line 77: define "intermediate"

Line 189-196: why are the relationships with the nutrient concentration/ratios not shown. Is a r^2 of 0.03 really significant?

Also N^* and Si^* are common parameters but are not always calculated in the same way in the community. They need to be defined in the text here.

Geibert, W., P. Assmy, D. C. E. Bakker, C. Hanfland, M. Hoppema, L. E. Pichevin, M. Schroder, J. N. Schwarz, I. Stimac, R. Usbeck and A. Webb (2010). "High productivity in an ice melting hot spot at the eastern boundary of the Weddell Gyre." *Global Biogeochemical Cycles* 24.

** Visit Nature Portfolio's author and referees' website at www.nature.com/authors for information about policies, services and author benefits**

Communications Earth & Environment is committed to improving transparency in authorship. As part of our efforts in this direction, we are now requesting that all authors identified as 'corresponding author' create and link their Open Researcher and Contributor Identifier (ORCID) with their account on the Manuscript Tracking System prior to acceptance. ORCID helps the scientific community achieve unambiguous attribution of all scholarly contributions. You can create and link your ORCID from the home page of the Manuscript Tracking System by clicking on 'Modify my Springer Nature account' and following the instructions in the link below. Please also inform all co-authors that they can add their ORCIDs to their accounts and that they must do so prior to acceptance.

If you experience problems in linking your ORCID, please contact the Platform Support Helpdesk.

Version 1:

Decision Letter:

Dear Ms Taylor,

Your manuscript titled "Giant iceberg behaviour impacts regional biogeochemical cycling in the Southern Ocean" has now been seen by our reviewers, whose comments appear below. In light of their advice we are delighted to say that we are happy, in principle, to publish a suitably revised version in *Communications Earth & Environment*.

We therefore invite you to revise your paper one last time to address the remaining concerns of our reviewers. At the same time we ask that you edit your manuscript to comply with our format requirements and to maximise the accessibility and therefore the impact of your work.

EDITORIAL REQUESTS:

****Please take care to match our formatting and policy requirements. We will check revised manuscript and return manuscripts that do not comply. Such requests will lead to delays. ****

SUBMISSION INFORMATION:

In order to accept your paper, we require the files listed at the end of the Editorial Requests Table; the list of required files is

also available at <https://www.nature.com/documents/commsj-file-checklist.pdf> .

OPEN ACCESS:

Communications Earth & Environment is a fully open access journal. Articles are made freely accessible on publication. For further information about article processing charges, open access funding, and advice and support from Nature Portfolio, please visit <https://www.nature.com/commsenv/open-access>

Link Redacted

Best regards,

Jose Luis Iriarte Machuca, PhD
Editorial Board Member
Communications Earth & Environment

Nicola Colombo, PhD
Associate Editor, Communications Earth & Environment
Consulting Editor, Communications Sustainability

REVIEWERS' COMMENTS:

Reviewer #1 (Remarks to the Author):

I have read through the very comprehensive rebuttal by the authors and read the manuscript, taking note of the changes made, as outlined in the rebuttal. I found the attention to the referees' points to be extremely positive and leading to a significantly more nuanced manuscript. I found no points that I wished to raise for attention.

Reviewer #2 (Remarks to the Author):

The manuscript titled "Giant iceberg behaviour impacts regional biogeochemical cycling in the Southern Ocean" examines the influence of giant icebergs (A-76A and A-23A) on biogeochemical cycling by altering local circulation and nutrient availability. The revised manuscript is a clear improvement, particularly in placing the varying effects of icebergs into context by discussing differences in background oceanic conditions.

The manuscript is well written, and the study provides an interesting discussion of the diverse effects of giant icebergs observed through opportunistic sampling.

The manuscript is recommended for publication following a few major modifications.

General

1. Figure clarity

As stated during the first review, the figures are aesthetically pleasing. However, there were significant issues with their interpretability. The authors have attempted to address these comments by updating the figures, but unfortunately, several problems remain.

a.
The authors respond that: "Variables that span positive and negative values, where the sign is scientifically meaningful, now use diverging colour scales with distinct hues on either side of zero." However, in Figures 2, 5, and 6, the diverging colour scales do not switch at zero. Instead, the colour transitions appear to be arbitrarily selected within the range spanned by the two icebergs. This is not necessarily problematic, but it is a misleading rebuttal reply. Additionally, in several instances it makes interpreting the data more difficult.

b.
The authors further state that “Colour scale limits have been adjusted to include additional buffering beyond the data extrema, reducing saturation effects and making it easier to infer relative values near the upper and lower bounds.” At least in the PDF provided to this reviewer, this does not appear to be correct. In multiple cases, the most extreme discrete measurements (i.e., individual data points) are darker or lighter than the bounds shown on the colour scale bar, which significantly limits interpretability.

Examples include:

1. Figures 2a and 2b: The transition between red and blue occurs at ~ 1 °C rather than zero (acceptable if intentional).
2. Figure 4a: The darkest point (the northernmost measurement) corresponds to the highest F_{met} fraction. According to line 248, this value should be 3.94%. However, the colour of this point (hex: 2D6A83) is darker than the upper limit of the colour scale (hex: 4090A8), suggesting an F_{met} fraction exceeding 5%.
3. Figures 5a and 5b: The transition between light and dark purple occurs at ~ 24 μM . This is both far from the zero where the authors described a color hue switch, and in a scale that is only positive. (No negative sign). Thus the rationale for a hue-switch is not apparent.
4. Figure 5c: The darkest nitrite measurement has a colour (hex: 821D49) darker than the upper bound of the colour scale (hex: AB5F79).
5. Figures 5e and 5f: The transition between light and dark blue occurs at ~ 2.4 μM . The rationale for this choice is unclear. Again, there are only positive values.
6. Figure 5g: The darkest DSi measurement (hex: 3D897F) is darker than the colour bar limit (hex: 449086), again implying more elevated values than described in the text.
7. Figures 6a and 6b: The transition between colour schemes occurs at ~ -3.8 . Additionally, the colour scheme is difficult to interpret when both extremes are rendered in shades of blue.
8. Figure 6d: The darkest green (hex: 33756C) falls outside the bounds of the colour bar (upper limit hex: 5B928A).

For these reasons, I must again recommend that the majority of figures be revised so that true interpretability is possible. The inconsistencies between the figures, the text, and the rebuttal responses reduce confidence in the synchrony between the manuscript and the visual presentation of data.

Additional Comments

- Lines 172-173: The text states: “Sea surface temperature (SST) was higher near A-76A (0.5 to 2.46 °C) than A-23A (-1.03 to -0.32 °C; $p < 0.001$; Figure 2).” However, in Figure 2b, (A-23A) all points are blue – suggesting they are temperatures above zero. Similarly, Figure 2a (A-76A) has all points as red, suggesting SST below zero. As noted in the first review, either the text or the figure is incorrect.
- Line 228: It would be clearer to refer to marine $\delta^{18}\text{O}$ or measured $\delta^{18}\text{O}$, rather than the only: “ $\delta^{18}\text{O}$ is sensitive...”.
- Lines 420-433: The broader impact of the study is only addressed in the final sentence of the manuscript. This section would benefit from a more explicit summary of the main conclusions and a clearer discussion of their broader implications.

Reviewer #3 (Remarks to the Author):

I am very happy with the new manuscript. The authors made a very good job of integrating all the reviewers' feedback in a coherent, well-written manuscript. As a result, the revised version is much stronger and presents a mechanistically sound and nuanced understanding of the impact of icebergs on SO productivity. I am now satisfied that it can be published in Nature Communications.

** Visit Nature Portfolio's author and referees' website at <http://www.nature.com/authors> for information about policies, services and author benefits**

Response to reviewer comments on “Giant iceberg behaviour impacts regional biogeochemical cycling in the Southern Ocean”

Communications earth and environment

Manuscript ID: COMMSENV-25-5511-T

Corresponding authors: Laura Taylor (lauror77@bas.ac.uk), Clara Manno (clanno@bas.ac.uk)

We thank the editors and reviewers for their constructive feedback, which has led to substantial improvements in clarity, interpretation, and contextualisation. The manuscript has been revised to present a more robust and fully supported explanation for the differences in productivity responses observed between A-76A and A-23A and to clearly articulate the implications of these findings for Southern Ocean biogeochemistry.

The abstract and introduction have been revised to explicitly convey the study's aim of understanding the mechanism controlling the magnitude and expression of primary productivity responses to giant iceberg passage. The text now clearly highlights that productivity enhancement results from the interaction of three factors: micronutrient fertilisation, which alleviates micronutrient (iron) limitation and enables growth; macronutrient resupply, which sustains and amplifies biomass accumulation; and the pre-existing physical and biogeochemical state of the surface ocean, which modulates the system's response to these perturbations. In support of this, the revised manuscript explicitly discusses the contrasting geographical locations of the two icebergs and the background environmental conditions at the time of sampling, including nutrient concentrations, physical oceanography, and pre-existing phytoplankton biomass.

The role of trace micronutrients, particularly iron, has been made explicit throughout providing a clear mechanistic basis for why iceberg passage has been proven to stimulate productivity in previous studies. References to $\delta^{30}\text{Si}$ have been reframed to avoid overstating its novelty, presenting it instead as a complementary tracer that, together with other biogeochemical observations, supports interpretation of nutrient limitation and biological response. The manuscript now integrates the two case studies to show how these factors combine: A-76A shows high-magnitude, sustained productivity linked to meltwater input and upwelling of nutrient-rich water, while A-23A exhibits a lack of response due to limited micronutrient input.

Collectively, these revisions provide a coherent and evidence-based explanation for the observed heterogeneity in iceberg-associated productivity enhancement and situate the findings within the broader context of Southern Ocean nutrient cycling and ecosystem responses.

During the revision process, it became clear that E Povl Abrahamsen has contributed substantially to the preparation and analysis of samples for this study, as well as providing important input during manuscript revision. To ensure appropriate attribution, E Povl Abrahamsen has been added as a co-author on the revised manuscript.

Response to reviewer 1

This paper examines nutrient uptake around two giant icebergs in the Weddell Sea in 2023. It finds very different behaviour in terms of many parameters, and particularly silicon and oxygen isotope fractionation, with the more recently calved A76A showing much more dynamic uptake than the older A23A. The paper is generally well written and clearly explained. The text and methods are well separated on the whole, and the supplementary figures are appropriate. I do not have major concerns about the paper and believe it is worthy of publication after a few points are expanded upon. These are explained below.

We would like to thank the reviewer for their constructive comments and are happy to address the recommendations below.

Point 1: While the paper clearly discusses the contrasting origins and life histories of the two icebergs I felt more might have been made of the decades difference in exposure to the ocean felt by the contrasting bergs. A23A had much more time to be changed by the ocean while it was grounded for decades, compared to the short ocean lifetime of A76A. How much might the difference in nutrient behaviour be explained by this exposure? For example, if sediments contained in the ice are more likely to be near the edges, much more of these would have been lost by A23A before it entered the ACC than in A76A's case. Is it possible to estimate this size change from satellite images, so as to speculate on this effect?

We thank the reviewer for this insightful comment, which highlights an important contrast in the ocean exposure histories of A-23A and A-76A and its potential implications for nutrient supply. We agree that the prolonged grounding of A-23A likely allowed for substantially greater modification by oceanic processes prior to our sampling, particularly through iceberg decay and associated sediment loss.

While this idea was briefly touched on in the originally submitted manuscript through speculation that processes leading to smaller biological responses around icebergs

originating from sectors B and C with longer transit distances to the ACC may also apply to icebergs grounded for extended periods prior to ACC transit, we agree that the manuscript would benefit from a more explicit and focused discussion of this issue. We have therefore added a new paragraph to the revised manuscript (lines 117-128) that directly addresses this point.

Using satellite imagery from as close as possible to the original time of calving, and again around the time the iceberg became grounded, we estimate the reduction in surface area of A-23A during its ~30 year grounding period. This estimate is then used as a basis to discuss the likely loss of sediment contained within the ice, particularly near iceberg margins, and how this prolonged exposure and sediment loss may have reduced the potential contribution of limiting micronutrients from A-23A relative to A-76A, which experienced a much shorter ocean lifetime prior to sampling. In addition to the written discussion, we have added a new figure to the supplementary material (S1) showing the satellite imagery used to estimate iceberg size and area loss.

While it is not possible to constrain sediment distribution within the iceberg, and therefore the precise magnitude or spatial pattern of sediment loss, this addition directly addresses the reviewer's suggestion and provides additional context for the observed differences in nutrient behaviour between the two icebergs.

Point 2: The origin and trajectory of the two icebergs probably contains some indication of each berg's depth. This is not discussed, but may be important for the upwelling tendency and behaviour exhibited during the cruises. The underwater shape may also be rather different, given the different ocean exposure times too. Some discussion of the impacts of these parameters on the nutrient uptake would be good.

We thank the reviewer for this comment, which raises an important consideration regarding iceberg draft, submerged geometry, and their potential influence on upwelling and nutrient supply. We agree that differences in iceberg submerged structure could affect upwelling dynamics and nutrient entrainment.

While the exact draft and underwater geometry of either iceberg were not measured during the cruises, we have added a new paragraph to the revised manuscript (lines 140-146) to explicitly acknowledge these uncertainties and to place our observations in the context of existing studies of giant icebergs. Specifically, we note that both A-23A and A-76A were likely sufficiently deep to penetrate below the mixed layer, given previous observations of Antarctic giant icebergs of a similar surface area (e.g. Lucas et al., 2025; Stephenson et al., 2011; Silva et al., 2006). Under this assumption, both icebergs would have had the potential to induce buoyant upwelling and entrainment of nutrient-rich Circumpolar Deep Water, providing a mechanism for macronutrient resupply to surface waters.

Point 3: Section 2.3: I could not find formal definitions of F_{met} and F_{sim} . This should be presented at first appearance.

We thank the reviewer for noticing this omission. We have now updated the sentence in which F_{met} and F_{sim} first appear (line 230) to provide explicit definitions to read

“ $\delta^{18}\text{O}$ is sensitive to polar precipitation and glacial melt, while sea ice processes primarily affect salinity, enabling quantification of the fractional contributions of meteoric water (F_{met}) and sea ice melt (F_{sim}) to surface freshwater (see Methods) [34,35].”

Point 4: section 3.6 Data availability This needs to be added. It currently just says “TBC”

We thank the reviewer for noticing this omission. We have now received the DOI for the published dataset, and have added the Data Availability statement accordingly.

Response to reviewer 2

The manuscript titled “Giant iceberg behavior impacts regional biogeochemical cycling in the Southern Ocean” delves into the influence of giant icebergs on biogeochemical cycling by altering local circulation and nutrient availability. Conducted in 2023, the study involved two British research vessels sampling around two large icebergs, A-76A and A-23A. These icebergs exhibited contrasting conditions, with one supporting meltwater-induced diatom blooms while the other showed no visible biological changes due to meltwater. The authors employed silicon isotope fractionation as a tracer for diatom biological uptake. They emphasize that the contrasting icebergs they measured illustrate the highly heterogeneous effects of giant iceberg transport and melt on local biogeochemistry, which is a combination of complex meltwater delivery and local circulation.

This study provides valuable measurements that illustrate the diverse effects of giant icebergs. As the authors rightly point out, the delivery and consequently the impact of giant icebergs will increase in the future.

Taylor et al. present as comprehensive a dataset as possible from opportunistic sampling and provide a valuable discussion that evaluates the diverse impacts of icebergs.

The manuscript is well-written, presenting observational results in an engaging manner to explore the impacts of giant icebergs on local biogeochemical cycling in the Southern Ocean. The data have been thoroughly analyzed, including a co-analysis of nutrients, including silicon and its isotopes. The authors present the data in visually appealing, albeit unclear, figures. With some modifications, I believe this paper will be a valuable contribution to the field.

The choice of the journal is appropriate, and this article is recommended for

publication. However, there are a few minor and some moderate modifications that should be made. The authors may find the general and specific comments, questions, and linguistic corrections listed below helpful.

We would like to thank the reviewer for their constructive comments, and are happy to address the recommendations below.

General

1. Figure clarity: Generally, the figures are aesthetically pleasing, but there are significant issues with their interpretability. Specifically, the color scale bars often don't match the spectrum of colors depicted in the figures. This makes it challenging to accurately interpret the range and values of the samples in hydrographic or biogeochemical variables.

We thank the reviewer for this detailed and constructive feedback on figure clarity and interpretability. We agree that clear and unambiguous colour scaling is essential for accurate interpretation of hydrographic and biogeochemical data, and we have therefore revised the figures extensively in response to these comments.

a. Figure 1: Can the outlines be in different colors to distinguish between icebergs, Antarctic/island outlines, and other features? Additionally, the iceberg track appears to cross South Georgia. How did it do that?

Thank you for noting the apparent crossing of South Georgia by iceberg A-76A. We can confirm that this did not occur. This artefact arises from a known issue in some US NSIDC iceberg tracking products, where occasional timestamp errors introduce spurious positional “jumps”. We have now replaced the original track with a corrected version that has been verified against satellite imagery, and the revised figure no longer shows a land crossing. In addition, outline colours have been updated to provide clearer visual distinction between icebergs, land outlines, and other geographic features.

b. Figure 2: It's difficult to see the variation in color in panel (b). Moreover, it seems that the colors shown in at least panel (a) and panel (d) aren't represented on the color scales. Panel (a) has more red colors, while panel (d) has quite green/yellow colors that aren't shown on the temperature and salinity color gradients. This makes it challenging to accurately interpret the figure.

c. Figure 4: Again, the colors in the figures are confusing. Why are there brown colors in panel (d)? Also, the colors in panel (a) are difficult to match to a percentage in Fmet. There are some medium brown colors that don't appear on the scale when there's an abrupt shift from dark brown/red to light grey/pink.

d. Figure 5: Once again, the colors in the figure don't match the scale bars. For example,

silicic acid shows much darker blues than on the scale bar. Darker nitrite colors aren't shown on the scale bar of the light orange gradient. And the medium colors in nitrate plots aren't shown on the scale bar of dark red vs light grey/peach.

e. Figure 6: Colors again.

While the original figures used established colour maps from the *cmocean* and *ColorBrewer* packages, and the colour bars did formally span the full data ranges, we recognise the reviewer's comments that the resulting figures were nonetheless difficult to interpret. In particular, we acknowledge that multi-hue colour maps can make it challenging to visually associate intermediate colours with specific values, especially when gradients transition rapidly between hues.

In response, we have taken a holistic approach to revising colour scales throughout the manuscript. All figures have now been updated using colour maps designed to improve interpretability, accessibility, and consistency. Specifically:

- Sequential variables are now shown using single-hue colour scales that vary monotonically from light to dark and remain interpretable in greyscale.
- Variables that span positive and negative values, where the sign is scientifically meaningful, now use diverging colour scales with distinct hues on either side of zero.
- Colour scale limits have been adjusted to include additional buffering beyond the data extrema, reducing saturation effects and making it easier to infer relative values near the upper and lower bounds.

These changes have been applied consistently across all figures using colour scale bars, and we believe they substantially improve the clarity and interpretability of the visualised data.

2. Framing of silicon isotope fractionation: The authors' abstract and conclusion assert that they demonstrate silicon isotopes are sensitive tracers of biological response. However, the use of silicon isotope fractionation to trace diatom growth is well-established, meaning the authors could be more modest in framing the novelty of their study. The paper's strengths lie in providing new observations of heterogeneous conditions around giant icebergs rather than introducing a novel technique.

We thank the reviewer for this comment and agree with their assessment. The use of silicon isotope fractionation as a tracer of diatom uptake is well established, and the original framing in the abstract and conclusion overstated the novelty of this aspect of the study.

In response, we have substantially revised both the abstract and conclusion to adopt a more accurate framing. References to silicon isotopes as a novel or uniquely sensitive

tracer of biological response have been removed, and the emphasis has been shifted towards the primary strength of the manuscript: the presentation of new, spatially resolved observations of heterogeneous physical, biogeochemical, and biological conditions in the vicinity of giant icebergs, and the implications these have for our understanding of phytoplankton bloom dynamics. Silicon isotope measurements are now framed as one component of a multi-parameter observational approach used to interpret biological responses, rather than a methodological advance in their own right.

3. Discussion of varying conditions: The abstract states: “These contrasting regimes reveal that the influence of icebergs on ocean biogeochemistry is highly heterogeneous, likely reflecting differences in meltwater delivery and local circulation.” However, differences are not limited to local circulation. Not mentioned is the differing background conditions such as background levels of macronutrients. Blooms are supported when a limiting nutrient is supplied. Iceberg influence then only causes blooms when nutrients that were limiting are supplied. If macronutrients were sufficient beforehand, adding more, for instance due to meltwater-driven upwelling, won’t stimulate further blooms. The differing background concentrations were discussed briefly. More discussion could be added to the manuscript to include not only varying iceberg delivery but also varying background conditions.

We thank the reviewer for this insightful comment and agree that the original framing placed too much emphasis on iceberg-driven processes alone, without sufficiently accounting for the role of background environmental and nutrient conditions in modulating biological responses. We agree that iceberg influence will only stimulate enhanced primary productivity when it alleviates pre-existing limitations, and that macronutrient resupply alone will not necessarily lead to bloom development if nutrients are already sufficient.

In response to this comment, and to the extensive related feedback from Reviewer 3, we have substantially revised the framing of the manuscript to more clearly and explicitly describe the nutrient processes governing productivity enhancement around giant icebergs. The revised manuscript is now structured around a single organising question: *what controls the magnitude and expression of primary productivity responses to giant iceberg passage in the Southern Ocean?*

Throughout the revised manuscript, we emphasise that iceberg-associated productivity enhancement reflects the combined influence of (i) micronutrient input, which alleviates iron limitation and enables phytoplankton growth, (ii) macronutrient resupply, which can sustain and amplify biomass accumulation (see lines 275-301 of the revised manuscript), and (iii) the pre-existing physical and biogeochemical state of the surface ocean, and the nature of the iceberg itself, which modulate how the system responds to these perturbations. A-76A is presented as a case study in which iceberg-induced upwelling of Circumpolar Deep Water contributes substantially to

macronutrient resupply, which supports sustained high-magnitude blooms once micronutrient limitation is alleviated. In contrast, A-23A represents a system in which one or more of the requirements for enhancement of primary productivity are absent, reflecting limited meltwater influence, relatively homogenous macronutrient fields, and minimal biological uptake, such that high-magnitude productivity enhancement is not expressed.

These conceptual changes are captured explicitly in the revised conclusion and are supported throughout the manuscript by integrated physical, chemical, and isotopic observations. We believe this revised framing directly addresses the reviewer's concern and provides a clearer and more mechanistic explanation for the observed heterogeneity in iceberg-associated biological responses.

4. Separating Freshwater Fractions: In lines 174-176, the authors describe sea-ice formation. Negative values of F_{sim} indeed indicate sea ice formation, but the most negative values of F_{sim} occur where F_{met} is high. This suggests that it's not real sea-ice formation but rather non-realistic endmembers causing the mixing model to produce positive or negative values when summed to 1. Furthermore, the authors mention sea ice melt and formation but then don't discuss them again. I agree that the majority of freshwater in the region is likely melt from the giant iceberg. However, perhaps more realistic endmembers could be chosen to better represent the system. If not, at least qualify what may be realistic.

We thank the reviewer for this careful reading of the freshwater fraction results. The three-endmember $\delta^{18}\text{O}$ -salinity mass balance is solved as a closed system, such that increases in one freshwater component necessarily reduce the contribution of the others (such that the total always equals 100 %), even when endmembers are well constrained. Consequently, covariance between F_{met} and F_{sim} does not, by itself, imply a physical coupling between meteoric input and sea ice processes, nor does it indicate unrealistic endmember selection.

Endmembers used in this study were drawn from Brown et al. (2014) and Meredith et al. (2017). Circumpolar Deep Water and sea ice endmembers were taken from Meredith et al. (2017), as these are well-defined for the region of our study. Meteoric endmembers were taken from Brown et al. (2014), as these were selected to reflect meteoric inputs to the Weddell Gyre, where glacial contributions are dominated by ice originating from the Filchner-Ronne Ice Shelf, which was the source of icebergs A-76A and A-23A. As stated in the Methods, the largest uncertainty in freshwater fraction determination arises from the mean meteoric water $\delta^{18}\text{O}$ endmember, which represents a combination of local, seasonally variable precipitation and glacial melt. Sensitivity analyses by Brown et al. (2014) indicate uncertainties in the derived freshwater

fractions of better than 1 % of the total sampled fluid volume, which remains applicable here.

To avoid over-interpretation, we have revised section 2.3 to clarify that the sea ice endmember is included to separate meteoric freshwater from other freshwater sources, rather than to diagnose sea ice melt or formation processes. The revised text now reads:

“Freshwater fractions are derived within a closed three-endmember mass balance, meaning that increases in one freshwater component necessarily reduce the contribution of one or more other endmembers. In this study, the quantification of sea ice melt is included to separate meteoric freshwater from other freshwater sources, rather than to investigate sea ice melt or formation processes in their own right (see S2).” (lines 231-236).

Consistent with this clarification, we have removed the previous interpretation of F_{sim} values (lines 174-176 in the original manuscript) and moved Figure 4 panels (c) and (d) to the Supplementary Material, as these are not central to the results presented here but may be of interest to readers considering the spatial distribution of freshwater endmembers.

While addressing this comment, we identified a minor typographical error in the $\delta^{18}O$ value used for the sea ice endmember. This has now been corrected in the analysis, and the corresponding text and figures. The correction results in only very small changes to the calculated freshwater fractions and does not affect the interpretation of the results. However, when considering F_{met} correlations with macronutrients and macronutrient tracers, this has resulted in a small shift in R^2 and p-values, such that the relationship between F_{met} and NO_3^- is now significant and has therefore been included in the discussion of these relationships (lines 316-319 in the revised manuscript).

5. Clarification of Si methodology: The methods are generally consistent with standard $\delta^{30}Si$ protocols. However, it would be helpful if the authors provided additional detail on the potential effects of the initial acidification step on dissolved Si integrity and recovery or confirmed that acidification did not introduce measurable Si loss prior to MAGIC processing.

Thank you for noting this concern around the lack of detail on the initial acidification step on dissolved Si integrity and recovery. The acidification step is primarily to get the pH seawater samples to between 5-7, which ensures optimum precipitation of brucite on addition of NaOH. However, the pH is not reduced to sufficiently low (< 2) that Si would polymerise, resulting in a loss of dissolved Si.

To reflect this, we have amended the text with the following:

“and to achieve a pH between 5-7 for subsequent processing.” (line 525)

“Si concentrations were checked, confirming all Si yields from the MAGIC method were above 90 %.” (lines 532-533)

Specific comments:

- *Lines 10-11: 10-20% of the SO carbon flux.*

Thank you for noting this from the cited paper. The text is now updated to read “By stimulating primary production and enhancing organic carbon export, they may account for 10 to 20 % of Southern Ocean carbon flux”. (lines 10-11).

- *Lines 121-122: Are the icebergs referenced backwards? It seems colder near A76-A not higher.*

The text as originally written is correct; however, we recognise that the apparent discrepancy likely arose from a difficulty interpreting the original figure colour scales. As described in our response to comments on figure clarity, we have now revised the colour scales throughout the manuscript to improve interpretability. In the updated Figure 2, there is now a diverging colour bar for Temperature, with distinct hues either side of zero to make clear the distinction between A-76A and A-23A.

- *Line 135: Clarify that it is in discrete samples.*

We thank the reviewer for noting the ambiguity in this statement. The algal standing stocks are not from discrete samples; in Figure 3, the points represent ship-board continuous underway chlorophyll- α measurements averaged over 5-minute intervals, placing values in discrete bins. To clarify this, we have updated the figure caption to explain the data source and processing more clearly.

Due to other revisions to the manuscript, the sentence as written in line 135 of the originally submitted manuscript is no longer present; however, the distinction between ship-board and remotely sensed chlorophyll- α data is described in lines 190-192 of the revised manuscript.

- *Line 170: Reference supplemental figures in order. Not S3 first.*

Thank you for noting this discrepancy. We have now reordered the supplementary material to reflect order of reference in the main text.

- *Additionally, S1 and S2 are not referenced in the manuscript text? Why are they then included?*

Thank you for noting this discrepancy. The appropriate reference to Figure S1 (now Figure S3) has been added to the text (line 309). Figure S2 has been removed from the Supplementary Material, as is not directly relevant to the text.

- *Lines 189-196: There is a very abrupt transition from discussing meltwater to biogeochemistry. This paragraph could use more introduction and explanation. Additionally, the relationship is discussed but not presented as a figure.*

We thank the reviewer for this suggestion. To improve the flow and contextual clarity, we have relocated discussion of F_{met} correlations to the macronutrients discussion (section 2.4, lines 316-319), where it now forms a more cohesive part of the narrative linking meltwater input to biogeochemical responses. In addition, Figure 7 in the revised manuscript has been added to explicitly illustrate the relationships discussed.

- *Line 191: N^* needs to be introduced.*

Thank you for noting this issue. N^* is now introduced on first appearance in line 303.

- *Line 206-207: In the figure it looks like the opposite where A-23A had higher N^* . Though it is difficult to tell due to the colors. (see figure comments above)*

We thank the reviewer for this observation. It is correct that N^* was significantly higher around A-23A than A-76A, as stated in the original manuscript. In the revised manuscript, it is now clearly indicated in the text (lines 305-306). As described above, the colour bars have been revised throughout the manuscript to improve interpretability.

- *Line 213: Reorder supplemental materials*

Thank you for noting the issue with supplemental material order. This has been ordered by appearance in the main text in the revised version.

- *Line 223: "...the range observed around A-23A". Could the authors provide more context. The range is not stated and the colors do not show the full spectrum so the full range cannot be assessed.*

Thank you for noting the lack of clarity in this sentence. The text has now been amended to read:

“High Si* values around both icebergs indicate that DSi is replete relative to nitrate and does not limit diatom growth [40]. Around A-23A, Si* ranged from 44.7 to 62.6 μM , largely consistent with upwelled CDW [44]. Around A-76A, Si* ranged from 25.3 to 62.9 μM ; the lower minimum may reflect partial DSi drawdown from upwelled CDW, potentially occurring at Si:N ratios $> 1:1$ under iron limitation [44,45].” (lines 329-333).

The figure colour bars have also been adjusted to reflect the full range, which should improve clarity.

- *Line 276-279: Could be shown in figures? Might be interesting.*

We thank the reviewer for this suggestion. The information referred to in lines 276-279 of the original manuscript (now lines 398-403), regarding the relationships of $\delta^{30}\text{Si}_{\text{DSi}}$ with NO_3^- and PO_4^{3-} , is now represented in Figure 8 (panels b, c, and d), allowing the pattern referenced to be more clearly visualised.

Response to reviewer 3

I have now reviewed the manuscript entitled “Giant iceberg behaviour impacts regional biogeochemical cycling in the Southern Ocean” by Taylor and co-authors. This study presents interesting physical, chemical and nutrient data around 2 icebergs and endeavours to better constrain the mechanisms and factors whereby Icebergs can enhance (or not) biological productivity in the Southern Ocean (SO).

Unfortunately, I find the paper to be unconvincing on several grounds (which I will detail below) and overall, I believe it lacks the scope and wide-ranging implications that are usually expected of publications in Nature Communications.

We thank the reviewer for their careful evaluation of our manuscript and for highlighting both the strengths and perceived limitations of the study. We have reflected carefully on the points raised and have made substantial revisions throughout the manuscript to improve clarity, strengthen the narrative, and more clearly articulate the broader implications of our observations for regional biogeochemical processes and productivity in the Southern Ocean. Specific changes addressing the reviewer’s detailed comments are described below, in addition to the opening summary of major changes.

1) While I agree with the “first-degree” interpretations of the data I find that the authors do not in fact provide new, clear or well-justified mechanisms to explain the difference in productivity responses seen between the two icebergs, which they announce they set off to do in the introduction. Or if they do, such mechanisms are not clearly explained in the text and hardly discussed at all in the later part of the manuscript especially in the context of the different geographical locations and background conditions of the

icebergs at the time of sampling, leaving it to the reader to infer what the factors that trigger these different responses actually are: is it melting that induces small scale upwelling of nutrients? The authors seem to point us in that direction. If yes, can the amount of melting be (semi)quantitatively linked to the amount of surplus productivity vs. background? Or is the main factor background temperatures which induces melting? Or is it background nutrient concentrations?... For example, it would have been very pertinent to show/measure background nutrient concentrations in the surface waters and compare it to the nutrient concentrations immediately around Iceberg A76 in particular (as this iceberg appears to impact productivity) in order to further ascertain whether there was indeed increased nutrient drawdown in the vicinity of this iceberg vs. in iceberg free zones. the background $d^{30}Si$ seem to corroborate this. As it stands, the main conclusion of the paper, although not expressly stated, appears to be: higher SSTs = more melting = more influence of the iceberg on the surface/subsurface water = more productivity. This is hardly new or unexpected.

We thank the reviewer for this detailed feedback. We agree that the original manuscript did not fully convey the mechanisms controlling the differing productivity responses around A-76A and A-23A. In response, we have substantially revised the manuscript to clarify these mechanisms throughout the text.

The revised manuscript now explicitly explores the geographical locations and background environmental conditions of the two icebergs, including nutrient concentrations, physical oceanography, and pre-existing phytoplankton biomass, and demonstrates how these factors interact with iceberg-driven processes to trigger and sustain productivity responses. Importantly, our study provides direct evidence of nutrient drawdown in the vicinity of giant icebergs, which allows us to distinguish whether elevated chlorophyll- α and low nutrient concentrations result from in situ biological uptake rather than lateral advection or simple dilution. This evidence was previously suggested by could not be demonstrated without the combined use of macronutrient measurements, satellite/in situ chlorophyll- α , and silicon isotopes.

We now clearly identify three interacting controls:

- (i) Micronutrient fertilisation, which alleviates iron or trace nutrient limitation and enables phytoplankton growth (as established in previous studies).
- (ii) macronutrient resupply, which sustains and amplifies biomass accumulation (supported by the observations in this study).
- (iii) background environmental preconditioning, which modulates the magnitude of the biological response (shown in previous work and emphasised here in the context of these specific icebergs).

These revisions make explicit that the productivity response is not solely a function of iceberg melting or sea surface temperature, but results from the dual-control mechanism of micronutrient stimulation and macronutrient resupply, moderated by local environmental context. By integrating these lines of evidence, the manuscript now presents a clear, well-supported, and mechanistic explanation for the contrasting responses between A-76A and A-23A.

It is clear from the data that around A76 (compared to A23) background temperatures are higher, making the melting more intense, the background nutrient concentration is lower while background phytoplankton presence is higher, resulting in the drawdown of Si and increased productivity in patches trailing off the iceberg. Yet it is also clear that A76, at the time of sampling was in the ACC / fronts region while A23 was located southward in a hydrologically and dynamically very different setting. Could it be that the differences seen between the productivity responses around A76 and A23 are simply or at least partly due to the background conditions in the 2 locations as opposed to iceberg diversity itself. What about stratification differences between both areas? What about background nutrient levels in the subsurface and surface? The potential influence of the oceanographic settings for the 2 icebergs on the productivity response is never properly discussed. Maybe it is irrelevant, but it should be mentioned/discussed.*

**by the way, the alleged intrinsic "diversity" of both icebergs -as per the abstract- is never really established by the authors in the text (how are they different?). In what way do these differences impact the productivity response? This is not explained.*

We thank the reviewer for these important points. We agree that, in the original manuscript, the influence of contrasting background oceanographic conditions and regional settings was not explored in sufficient depth, which limited the clarity of mechanistic interpretation. In response, we have substantial revised sections 2.1 (Trajectory and evolution of A-76A and A-23A) and 2.2 (Environmental contexts shape iceberg biogeochemical influence) to more explicitly address how background physical, chemical, and biological conditions modulate productivity responses to iceberg passage.

The revised manuscript clarifies that differences in productivity between A-76A and A-23A cannot be attributed to iceberg-related processes alone, but instead arise from the interaction between iceberg-derived perturbations and the pre-existing environmental state. The differing regional settings of A-76A and A-23A are now clearly described in the context of their sampling locations and trajectories (lines 96-116), including A-76A's extended residence in the ACC frontal zone (lines 102-107) and A-23A's decades-long grounding in the Weddell Sea (lines 117-128). Differences in surface hydrographic properties (SST and salinity) and their relationship to regional climatology are explicitly

discussed (lines 162-183), alongside examining how chlorophyll- α concentrations compare to those typical in each region (lines 194-198). These elements are subsequently integrated into the interpretation of macronutrient distributions, nutrient drawdown, and productivity responses.

In particular, we now emphasise that A-76A transited a dynamically active ACC frontal environment characterised by higher background temperatures, elevated phytoplankton biomass, and likely depleted surface macronutrients, creating conditions under which iceberg-induced micronutrient fertilisation and macronutrient resupply could stimulate and sustain enhanced primary production. In contrast, A-23A was sampled in a colder, more weakly productive region with distinct hydrographic structure and background conditions, that may have limited, though not necessarily precluded, the expression of any iceberg-driven productivity responses.

We also agree that the term “iceberg diversity” in the original manuscript was insufficiently defined. In the revised text, we now explicitly describe how differences in iceberg age, grounding history, residence time, and cumulative environmental exposure may influence sediment retention, micronutrient delivery, and interaction with subsurface water masses (lines 147-155). These characteristics are now clearly linked to their potential biogeochemical consequences, rather than being treated as intrinsic but undefined properties.

Overall, the revised manuscript explicitly recognises background environmental conditions as a key control on iceberg-associated productivity responses and demonstrates that the contrasting responses observed around A-76A and A-23A reflect the combined influence of iceberg history and regional oceanographic context.

2) There are strong indications that some icebergs can deliver Iron to the surface ocean as they melt. (Geibert, Assmy et al. 2010) published about this mechanism specifically in the Weddell Sea. Today the SO is an HNLC region where productivity is vastly limited by the lack of Fe. While iron is vaguely mentioned in the text, the possibility of its influence on the productivity responses is not explored. This to me is an important omission. Iron limitation is inferred from N^ by the authors but the rationale behind this is not well justified. Are there Iron data available to the authors? Could the icebergs, coming from different areas and calving at different time have a different iron load?*

We thank the reviewer for raising this important point. We agree that while the role of trace micronutrients, particularly iron, was implied in the original manuscript, it did not sufficiently articulate this role or integrate it into interpretation of the findings of this study.

In the revised manuscript, we have substantially reframed the interpretation to centre limiting nutrient fertilisation as a primary control on iceberg-induced productivity enhancement, while clearly disentangling this process from the role of macronutrient resupply. Specifically, we now describe iceberg-associated productivity responses as emerging from the interaction of three processes: micronutrient fertilisation, macronutrient resupply, and background environmental preconditioning.

To make the importance of micronutrient fertilisation explicit throughout the manuscript, we have expanded the discussion of iron limitation and iceberg-derived micronutrient supply in several locations. These include the introduction where iron fertilisation by icebergs (lines 35-50) is positioned as a key underlying mechanism integral to the framing of this study (lines 65-86); the description of A-76A's extended residence in the ACC frontal zone and its implications for sustained nutrient input (lines 102-107); discussion of how A-23A's prolonged grounding may have resulted in sediment loss and limited micronutrient fertilisation (lines 121-128); explicit discussion of the mechanisms by which icebergs alleviate nutrient limitation (lines 206-213); the entrainment of sediment-bound micronutrients into glacial meltwater (lines 256-257); the role of micronutrient supply in enabling macronutrient resupply to translate into sustained productivity enhancement (lines 287-295); and discussion of how Si isotopes show the role of upwelling resupply of macronutrients as a component of productivity enhancement (lines 404-417).

We agree that iron limitation should not be inferred simplistically. In the revised text, we clarify that Si* may indicate patterns of nutrient utilisation that are consistent with iron limitation, but does not constitute a direct proxy for iron availability.

Unfortunately, no measurements of iron are available from these cruises. We believe the interpretation of the role iceberg-derived micronutrient play in enabling primary production enhancement is consistent with prior work in the region. We thank the reviewer for drawing attention to Geibert et al. (2010), which has now been incorporated into the discussion of previous studies (line 39).

The possibility that icebergs originating from different source regions with different calving histories carry differing sediment and iron loads is explicitly discussed in the introduction (lines 41-46) and revisited in the context of A-23A's long grounding and potential sediment depletion (lines 117-128).

3) I find the claim (introduction) that $d^{30}\text{Si}$ offers a unique tool to look at the variable productivity responses to icebergs to be overblown as it would not show anything different from $d^{15}\text{N}$ for instance. If the authors had used $d^{30}\text{Si}$ in conjunction with $d^{15}\text{N}$ for instance, the iron limitation status in the surface water could have been established/inferred and this would have been a more powerful diagnostic tool.

We thank the reviewer for this constructive comment and agree that the original phrasing in the Introduction overstated the diagnostic uniqueness of $\delta^{30}\text{Si}$. We acknowledge that silicon isotopes do not provide fundamentally different information from other nutrient isotope systems, such as $\delta^{15}\text{N}$, when considered in isolation in this context, and that $\delta^{30}\text{Si}$ alone cannot be used to establish iron limitation status in the surface waters.

In response, we have revised the Introduction and Abstract to remove claims that $\delta^{30}\text{Si}$ offers a unique tool for diagnosing variable productivity responses to icebergs. Instead, $\delta^{30}\text{Si}$ is now framed more modestly as one component of a multi-parameter observational approach, consistent with the revised framing throughout the manuscript.

We retain $\delta^{30}\text{Si}$ as a valuable tracer in this study because diatoms dominate the observed productivity responses, and silicic acid drawdown is a key feature of the biogeochemical signal associated with iceberg passage. In this context, $\delta^{30}\text{Si}$ provides process-relevant information on diatom nutrient utilisation that complements nutrient concentrations, chlorophyll- σ , and hydrographic observations, without being presented as a standalone or uniquely diagnostic factor.

We believe this revised framing more accurately reflects both the established role of silicon isotopes in marine biogeochemistry and their appropriate contribution to the interpretation of iceberg-associated productivity responses in the study.

4) The closing paragraphs remain vague and do not offer the clear mechanisms promised in the introduction. The implications of the findings (which are unclear) are not well-discussed.

We thank the reviewer for this comment and acknowledge that, in the original manuscript, the concluding paragraphs did not sufficiently distil the mechanistic insights developed throughout the manuscript or clearly articulate their broader implications.

In response, the Conclusion has now been substantially revised to provide a clear and explicit synthesis of the mechanisms resolved by our observations. The revised text now states that sustained high-magnitude productivity enhancement from giant iceberg passage arises from a dual-control mechanism, in which micronutrient input from iceberg meltwater initiates phytoplankton growth, while physically driven macronutrient resupply, via iceberg-induced upwelling, sustains elevated productivity once growth is enabled. This mechanism is explicitly described for A-76A, using meteoric water fractions, macronutrient distributions, and silicon isotopic evidence, and contrasted with A-23A which does not have an observable meltwater influence,

and may have outer sediment, preventing the initiation of productivity enhancement by micronutrient input.

The revised conclusion clarifies that iceberg impacts of surface biogeochemistry are not uniform, but depends on the interaction between melt-driven freshwater input, nutrient entrainment, iceberg history, and background ocean structure. This framing directly addresses the contrasting responses observed between the two icebergs and provides a mechanistic explanation for why high-magnitude productivity enhancement is expressed in some settings but not others.

Finally, the broader implications of these findings are now explicitly discussed, highlighting the importance of incorporating this dual-control framework into predictive models to better constrain how increasing iceberg discharge under continued Antarctic ice sheet retreat may influence Southern Ocean biogeochemistry and carbon cycling (lines 430-433).

We believe these revisions address the reviewer's concern by ensuring that the conclusion delivers the clear mechanistic synthesis and implications that are motivated in the Introduction.

Minor comments:

Line 77: define "intermediate"

We thank the reviewer for highlighting this ambiguity. We have clarified the term by specifying the stage of the iceberg lifecycle referred to. The sentence now reads:

"Both observations were made during intermediate stages of the icebergs' lifetimes, after initial calving but prior to advanced fragmentation or decay." (lines 94-95).

Line 189-196: why are the relationships with the nutrient concentration/ratios not shown. Is a r2 of 0.03 really significant? Also N and Si* are common parameters but are not always calculated in the same way in the community. They need to be defined in the text here.*

We thank the reviewer for raising these points.

Clear definitions of N* and Si* have now been added to the text, which now reads:

"To investigate differences in nutrient supply and utilisation, we examined the stoichiometric tracers N* and Si*, which were calculated following commonly used formulations. Specifically, N* was calculated as $[\text{NO}_3^-] - 16[\text{PO}_4^{3-}]$ (following [39]) and Si* as $[\text{Si}(\text{OH})_4] - [\text{NO}_3^-]$ (following [40])." (lines 302-305).

To provide full visibility, we have now added a supplementary table (Table S1) detailing all relationships between F_{met} and nutrient concentrations and ratios. Significant relationships are also now shown in Figure 7 as per the request of Reviewer 2.

The previously reported R^2 value of 0.03 was a typographical error. The correct R^2 is 0.32, which is now reported in the text (line 319).

Response to reviewer comments on “Giant iceberg behaviour impacts regional biogeochemical cycling in the Southern Ocean”: 09.03.2026

Communications earth and environment

Manuscript ID: COMMSENV-25-5511-T

Corresponding authors: Laura Taylor (lauror77@bas.ac.uk), Clara Manno (clanno@bas.ac.uk)

We thank the Reviewers and Editors for the time and care taken to evaluate this manuscript. We are delighted that the manuscript has been accepted for publication following these final revisions.

Response to reviewer 1

I have read through the very comprehensive rebuttal by the authors and read the manuscript, taking note of the changes made, as outlined in the rebuttal. I found the attention to the referees' points to be extremely positive and leading to a significantly more nuanced manuscript. I found no points that I wished to raise for attention.

We thank the reviewer for their thoughtful and constructive feedback on the manuscript. Their comments helped us improve the clarity and nuance of the work, and we appreciate their careful evaluation during the review process.

Response to reviewer 2

The manuscript titled “Giant iceberg behaviour impacts regional biogeochemical cycling in the Southern Ocean” examines the influence of giant icebergs (A-76A and A-23A) on biogeochemical cycling by altering local circulation and nutrient availability. The revised manuscript is a clear improvement, particularly in placing the varying effects of icebergs into context by discussing differences in background oceanic conditions.

The manuscript is well written, and the study provides an interesting discussion of the diverse effects of giant icebergs observed through opportunistic sampling.

The manuscript is recommended for publication following a few major modifications.

We thank the reviewer for their constructive comments and positive assessment of the revised manuscript. We address the individual points raised below.

General

1. Figure clarity

As stated during the first review, the figures are aesthetically pleasing. However, there were significant issues with their interpretability. The authors have attempted to address these comments by updating the figures, but unfortunately, several problems remain.

a.

The authors respond that: “Variables that span positive and negative values, where the sign is scientifically meaningful, now use diverging colour scales with distinct hues on either side of zero.” However, in Figures 2, 5, and 6, the diverging colour scales do not switch at zero. Instead, the colour transitions appear to be arbitrarily selected within the range spanned by the two icebergs. This is not necessarily problematic, but it is a misleading rebuttal reply. Additionally, in several instances it makes interpreting the data more difficult.

b.

The authors further state that “Colour scale limits have been adjusted to include additional buffering beyond the data extrema, reducing saturation effects and making it easier to infer relative values near the upper and lower bounds.” At least in the PDF provided to this reviewer, this does not appear to be correct. In multiple cases, the most extreme discrete measurements (i.e., individual data points) are darker or lighter than the bounds shown on the colour scale bar, which significantly limits interpretability.

Examples include:

1. Figures 2a and 2b: The transition between red and blue occurs at ~ 1 °C rather than zero (acceptable if intentional).
2. Figure 4a: The darkest point (the northernmost measurement) corresponds to the highest F_{met} fraction. According to line 248, this value should be 3.94%. However, the colour of this point (hex: 2D6A83) is darker than the upper limit of the colour scale (hex: 4090A8), suggesting an F_{met} fraction exceeding 5%.
3. Figures 5a and 5b: The transition between light and dark purple occurs at ~ 24 μ M. This is both far from the zero where the authors described a color hue switch, and in a scale that is only positive. (No negative sign). Thus the rationale for a hue-switch is not apparent.
4. Figure 5c: The darkest nitrite measurement has a colour (hex: 821D49) darker than the upper bound of the colour scale (hex: AB5F79).
5. Figures 5e and 5f: The transition between light and dark blue occurs at ~ 2.4 μ M. The rationale for this choice is unclear. Again, there are only positive values.
6. Figure 5g: The darkest DSi measurement (hex: 3D897F) is darker than the colour bar limit (hex: 449086), again implying more elevated values than described in the text.
7. Figures 6a and 6b: The transition between colour schemes occurs at ~ -3.8 . Additionally, the colour scheme is difficult to interpret when both extremes are rendered in shades of blue.
8. Figure 6d: The darkest green (hex: 33756C) falls outside the bounds of the colour bar (upper limit hex: 5B928A).

For these reasons, I must again recommend that the majority of figures be revised so that true interpretability is possible. The inconsistencies between the figures, the text, and the rebuttal responses reduce confidence in the synchrony between the manuscript and the visual presentation of data.

We thank the reviewer for again raising concerns regarding figure interpretability. After carefully considering these comments, we investigated whether the discrepancies might arise from differences in how the figures are rendered in different PDF viewers.

During this investigation, we observed that the colour bars appear differently when the manuscript PDF is viewed using the built-in Mac Preview application compared with other viewers such as Adobe Acrobat, Google Chrome, or Microsoft Edge. When viewed in Mac Preview, the colour scales are noticeably altered relative to both the original exported .jpg figures and the rendering seen in other PDF viewers. These changes include compression of the colour scale range, shifts in colour transitions, more abrupt colour transitions, and complete reversal of colours. These issues appear consistent with the issues described by the Reviewer.

Because we are not aware of which software was used to view the manuscript PDF, we below provide a side-by-side comparison illustrating (i) the original .jpg figure as exported for the manuscript, and (ii) the appearance of the same figure when viewed using Mac Preview. In the original .jpg files, the colour scale limits and transitions correspond correctly to the values described in the text, and the issues highlighted by the reviewer are not present.

We therefore believe that the discrepancies noted arise from rendering differences in certain PDF viewers rather than from the figures themselves. These artefacts should not persist in the final production version of the manuscript.

Figure	Original .jpg	Mac Preview screenshot
1		2		 Fig. 2 Temperature and salinity in the vicinity of giant icebergs A-76A and A-23A. Panels show temperature around A-76A (a) and A-23A (b) and salinity around A-76A (c) and A-23A (d).

3

Fig. 3 Satellite and ship-board chlorophyll-*a* concentrations surrounding giant icebergs A-76A and A-23A. Eight-day mean satellite chl-*a* concentration (mg m^{-3}) from Aqua-MODIS (4 km resolution; [34]) surrounding icebergs A-76A (a; 25 January – 1 February 2023), and A-23A (b; 3-10 December 2023). Discrete points show ship-board continuous underway chl-*a* concentrations averaged to 5-minute intervals.

4

Fig. 4 Meteoric water fraction distributions surrounding giant icebergs A-76A and A-23A. Panels show the meteoric water fraction (F_{mwt}) in the vicinity of icebergs A-76A (a) and A-23A (b).

5

Fig. 5 Macronutrient concentrations surrounding giant icebergs A-76A and A-23A. Panels show nitrate (a, b), nitrite (c, d), phosphate (e, f), and dissolved silicic acid (g, h) around A-76A (left) and A-23A (right), respectively. All concentrations are in μM .

6

Fig. 6 Nutrient deviations surrounding giant icebergs A-76A and A-23A. Panels (a) and (b) show N* (μM), while panels (c) and (d) show Si* (μM) around A-76A (left) and A-23A (right), respectively.

7

Fig. 7 Relationship between N* and meteoric water fraction around giant icebergs A-76A and A-23A. The relationship between N* and F_{met} is shown for waters surrounding icebergs A-76A (green) and A-23A (orange). Linear regressions are fitted for each iceberg, with corresponding equations, coefficients of determination (R^2), and p -values.

8

Fig. 8 Relationships between macronutrient concentration and silicon isotope composition and in waters surrounding giant icebergs A-76A and A-23A. Panel (a) shows the relationship between DSi concentration (μM) and $\delta^{30}\text{Si}_{\text{DSi}}$ (‰) for waters surrounding icebergs A-76A (green) and A-23A (orange), and initial CDW waters (black) (a). Panels (b-d) show the relationship between $\delta^{30}\text{Si}_{\text{DSi}}$ and DSi (b), nitrate (c), and phosphate (d). Linear regressions are fitted for each iceberg, with corresponding equations, coefficients of determination (R^2), and p -values. Error bars represent ± 0.100 ‰, equivalent to twice the maximum standard deviation of the reference standards.

9

Fig. 9 Dissolved silicon isotope composition surrounding giant icebergs A-76A and A-23A. Panels show $\delta^{30}\text{Si}_{\text{DSi}}$ (‰) in the vicinity of icebergs A-76A (a) and A-23A (b).

Additional Comments

• Lines 172-173: The text states: “Sea surface temperature (SST) was higher near A-76A (0.5 to 2.46 °C) than A-23A (−1.03 to −0.32 °C; $p < 0.001$; Figure 2).” However, in Figure 2b, (A-23A) all points are blue – suggesting they are temperatures above zero. Similarly, Figure 2a (A-76A) has all points as red, suggesting SST below zero. As noted in the first review, either the text or the figure is incorrect.

We thank the reviewer for highlighting this point. As shown in the correctly rendered figures above, positive SST values are displayed in shades of red and negative values in shades of blue, consistent with the text description. In the rendering observed in Mac Preview however, the colour scheme appears reversed, which likely explains the discrepancy noted by the reviewer.

- Line 228: It would be clearer to refer to marine $\delta^{18}\text{O}$ or measured $\delta^{18}\text{O}$, rather than the only: “ $\delta^{18}\text{O}$ is sensitive...”.

We thank the reviewer for this suggestion. This wording has been clarified in the revised manuscript.

- Lines 420-433: The broader impact of the study is only addressed in the final sentence of the manuscript. This section would benefit from a more explicit summary of the main conclusions and a clearer discussion of their broader implications.

We thank the reviewer for this helpful suggestion. In response, we have added a more explicit Conclusions section.

Response to reviewer 3

I am very happy with the new manuscript. The authors made a very good job of integrating all the reviewers' feedback in a coherent, well-written manuscript. As a result, the revised version is much stronger and presents a mechanistically sound and nuanced understanding of the impact of icebergs on SO productivity. I am now satisfied that it can be published in Nature Communications.

We thank the reviewer for their thorough and constructive comments during the review process. Their feedback was extremely helpful in strengthening the manuscript, and we are pleased that they are satisfied with the revised version.